# A feedback loop governs the relationship between lipid metabolism and longevity

**Nicole K Littlejohn[1], Nicolas Seban[1], Chung-Chih Liu[2], Supriya Srinivasan[1]***

[1]Department of Neuroscience and The Dorris Neuroscience Center, The Scripps Research Institute, La Jolla, United States; [2]Skaggs Graduate School of Chemical and Biological Sciences, The Scripps Research Institute, La Jolla, United States

**Abstract** The relationship between lipid metabolism and longevity remains unclear. Although fat oxidation is essential for weight loss, whether it remains beneficial when sustained for long periods, and the extent to which it may attenuate or augment lifespan remain important unanswered questions. Here, we develop an experimental handle in the *Caenorhabditis elegans* model system, in which we uncover the mechanisms that connect long-term fat oxidation with longevity. We find that sustained β-oxidation via activation of the conserved triglyceride lipase ATGL-1, triggers a feedback transcriptional loop that involves the mito-nuclear transcription factor ATFS-1, and a previously unknown and highly conserved repressor of ATGL-1 called HLH-11/AP4. This feedback loop orchestrates the dual control of fat oxidation and lifespan, and shields the organism from life-shortening mitochondrial stress in the face of continuous fat oxidation. Thus, we uncover one mechanism by which fat oxidation can be sustained for long periods without deleterious effects on longevity.

## Introduction

The relationship between lipid metabolism, adiposity and lifespan remains unclear. Support for this assertion comes from several sources. On the one hand, accumulation of lipid deposits in ectopic tissues is associated with age-associated illnesses including diabetes, hepatic and pancreatic steatoses and other metabolic illnesses that shorten lifespan (*Conte et al., 2019a*; *Schmeisser et al., 2019*; *Shulman, 2014*). Converting the excess lipid deposits to energy via a cascade of metabolic reactions: lipolysis of triglyceride lipids, fatty acid breakdown via beta-oxidation and increased electron transport chain activity in the mitochondria, have been proposed to benefit health and longevity (*Bonawitz et al., 2007*; *Conte et al., 2019b*; *Vatner et al., 2018*). In this context, *increased* metabolic activity and mitochondrial respiration increase lifespan (*Speakman et al., 2004*). On the other hand, long-term caloric restriction and decreased metabolic rate reduce electron transport chain activity in the mitochondria and increase lifespan (*Burkewitz et al., 2015*; *Durieux et al., 2011*; *Mattison et al., 2017*). This paradoxical observation, first made in rats (*McCay et al., 1935*) and then extended to several species including *C. elegans* (*Kimura et al., 1997*; *Lakowski and Hekimi, 1998*) and primates (*Mattison et al., 2017*; *Redman and Ravussin, 2011*), argues for the opposite: that *decreased* metabolic and mitochondrial activity increases longevity.

Additional data supporting a non-linear relationship between lipid metabolism, fatty acid oxidation and longevity come from studies of the systemic insulin and TGF-beta signaling pathways, which are master regulators of physiology in *Caenorhabditis elegans.* Loss of the insulin/insulin-like growth factor (IGF-1) receptor *daf-2*, or of the transforming growth factor-β (TGFβ) signal *daf-7*, each result in substantially increased adiposity and body fat stores, but with a concomitant increase in lifespan (*Greer et al., 2008*; *Kimura et al., 1997*; *Ogg et al., 1997*). In the case of the DAF-2 insulin pathway, there is a clear dissociation between fat storage and longevity amongst the various *daf-2* alleles (*Perez and Van Gilst, 2008*). This non-linearity between adiposity and longevity is echoed in human

*For correspondence:
supriya@scripps.edu

**Competing interests:** The authors declare that no competing interests exist.

observational studies that repeatedly suggest either no relationship (*Kuk and Ardern, 2009*; *Reynolds et al., 2005*), or an inverse association between adiposity and mortality in late life (*Fontaine et al., 2003*; *Stevens et al., 1998*; *Zheng and Dirlam, 2016*). A resolution to this dichotomy might stem from considering the nature of experimental interventions. The majority of *C. elegans* studies showing a relationship between reduced mitochondrial function and increased longevity come from manipulations that decrease fat oxidation and metabolic rate (*Dillin et al., 2002*; *Durieux et al., 2011*; *Lee et al., 2003*). However, the converse has not been tested. In other words, the consequences of a sustained increase in fat oxidation on longevity in non-disease states has remained unexplored. Furthermore, fundamental mechanisms that connect lipid metabolism with longevity regulation at an organismal level still remain poorly understood. To this end, we considered it fruitful to conduct an investigation into the effects of sustained fat loss and increased mitochondrial respiration on longevity.

The neuromodulator serotonin (5-hydroxytryptamine; 5-HT) is a major regulator of metabolism, behavior, and physiology in many species. In *C. elegans*, 5-HT is synthesized by the rate-limiting enzyme tryptophan hydroxylase (TPH-1) in only a few head neurons (*Sze et al., 2000*) but has wide-ranging effects across the organism including lipid metabolism (*Srinivasan et al., 2008*), behavioral responses to food (*Cunningham et al., 2012*), reproduction (*Tanis et al., 2008*), and pathogen avoidance (*Zhang et al., 2005*). 5-HT is a powerful stimulator of fat loss in the intestine (*Noble et al., 2013*), wherein the majority of lipids are stored and metabolized (*Srinivasan, 2015*). 5-HT-elicited fat loss occurs by activating the mitochondrial beta-oxidation pathway (*Srinivasan et al., 2008*), in which stored triglyceride lipids are oxidized to usable energy (*Salway, 1999*). Thus, approaches that increase neuronal 5-HT would have the potential to serve as a system to test the effects of sustained fat oxidation on lifespan. A major caveat is that genetic or pharmacological approaches that globally augment or decrease serotonin signaling lack specificity, leading to confounding and counterregulatory behavioral and metabolic effects that become difficult to disentangle. In this regard, an experimental approach that allows the selective manipulation of metabolic rate via increased fat oxidation would represent a valuable methodological advance.

We had recently defined a role for a tachykinin peptide called FLP-7 that couples neuronal signaling with fat oxidation in the intestine. FLP-7 is the *C. elegans* ortholog of the mammalian family of tachykinin peptides and is released from secretory neurons in response to fluctuations in neuronal 5-HTergic signaling. We discovered FLP-7 in the context of deciphering the neural circuit for 5-HT-mediated fat mobilization. In recognizing that the *C. elegans* gut is not directly innervated but that 5-HT, a neural signal, exerts profound effects on lipid metabolism in the intestine, we had sought to define the neuroendocrine mechanisms that allow communication from the nervous system to the gut. A genetic screen followed by molecular genetic analyses identified FLP-7 as the causative neuroendocrine secreted factor (*Palamiuc et al., 2017*). FLP-7 is secreted from the ASI neurons in which it is necessary and sufficient, and activates the G-protein-coupled receptor NPR-22 (ortholog of mammalian tachykinin receptor NK2R), in the intestinal cells (*Palamiuc et al., 2017*). FLP-7/NPR-22 signaling transcriptionally activates the enzyme adipocyte triglyceride lipase (ATGL; ATGL-1 in *C. elegans*), a highly conserved lipolytic enzyme that regulates the first step in the fat oxidation and energy production cascade: the hydrolysis of triglycerides to free fatty acids.

The tachykinin signaling pathway thus defines the core neuroendocrine axis by which the metabolic effects of neuronal 5-HT signaling are relayed to the intestine. Critically, we previously showed that FLP-7/NPR-22 signaling does not alter behaviors or physiological outputs associated with global 5-HT signaling (*Palamiuc et al., 2017*). Here, we leverage this knowledge and generate a *C. elegans* transgenic line that expresses the FLP-7 tachykinin peptide from the ASI neurons to specifically and selectively recapitulate all the effects of neuronal 5-HT on driving fat oxidation without concomitant behavioral effects. In principle, the FLP-7 transgenic line represents a potent experimental tool for us to address, in a non-disease context, the question of whether neuronally driven sustained fat oxidation has an effect on longevity.

## Results

### An experimental handle to generate sustained fat oxidation in the intestine

The FLP-7/NPR-22 tachykinin neuron-to-intestine signaling pathway is triggered by increases in neuronal 5-HT (*Figure 1A–C*), as previously published (*Palamiuc et al., 2017*). We tested several measures to ascertain the extent to which the integrated FLP-7 transgenic line (henceforth *flp-7$^{tg}$*) recapitulates the effects of neuronal 5-HT signaling on metabolic parameters in the intestine. First, we observed that *flp-7$^{tg}$* worms have a significant reduction in intestinal fat stores (*Figure 1D,E*), recapitulating that seen with 5-HT treatment (*Figure 1F,G*). The fat reduction elicited by *flp-7$^{tg}$* is dependent on the presence of the intestinal triglyceride lipase ATGL-1 (*Figure 1D,E*; note that the *atgl-1* null mutant is inviable), as was noted for 5-HT itself (*Figure 1F,G*; *Noble et al., 2013*; *Palamiuc et al., 2017*). In previous work, we uncovered major components of the mitochondrial beta-oxidation pathway that functionally connect ATGL-1 activity to the electron transport chain (ETC) that underlies mitochondrial respiration (*Noble et al., 2013*; *Srinivasan et al., 2008*). Accordingly, FLP-7-stimulated fat loss is accompanied by increased basal and maximal respiration that are both abrogated upon RNAi-mediated inactivation of *atgl-1* (*Figure 1H,I*), thus increased mitochondrial respiration in *flp-7$^{tg}$* animals results from conversion of stored triglycerides to energy via beta-oxidation and ETC activity. Finally, other behaviors elicited by 5-HT signaling including feeding rate (*Figure 1J*), locomotion on and off food (*Figure 1K*) and egg-laying (*Figure 1L*) remained unaltered in *flp-7$^{tg}$* animals. Thus, the *flp-7$^{tg}$* line fully recapitulates the metabolic effects of genetic and pharmacological manipulation of neuronal 5-HT signaling without altering other 5-HT-mediated behaviors (*Horvitz et al., 1982*; *Loer and Kenyon, 1993*; *Palamiuc et al., 2017*; *Song and Avery, 2012*; *Sze et al., 2000*; *Waggoner et al., 1998*) and can serve as a valuable experimental handle to examine the long-term effects of sustained fat oxidation and increased ETC activity.

### Intestinal fat oxidation via ATGL-1 induction evokes a mitochondrial stress response

Several lines of evidence have suggested that mild reductions in mitochondrial respiration can lead to a sustained increase in lifespan via decreased ROS production, hormesis, or other mechanisms (*Braeckman et al., 1999*; *Copeland et al., 2009*; *Dell'agnello et al., 2007*; *Durieux et al., 2011*; *Feng et al., 2001*; *Yang and Hekimi, 2010*). We examined *flp-7$^{tg}$* animals to test whether the observed increase in mitochondrial respiration (*Figure 1H,I*) might evoke the opposite effect on longevity. However, we found that *flp-7$^{tg}$* animals had nearly the same lifespan as wild-type animals, with a non-significant difference (p=0.08 by Log-rank Test) in survival probability statistics (*Figure 2A,B*). Next, we considered whether the increased mitochondrial respiration (*Figure 1H,I*) might evoke a systemic stress response. To test this possibility, we examined the effects of 5-HT administration or FLP-7 secretion on a variety of well-established stress reporters in *C. elegans* including those involved in the cytoplasmic heat-shock response (*hsp-70* and *hsp-16.2*), oxidative stress (*sod-3*), nutrient stress (DAF-16 nuclear localization), ER stress (*hsp-4*) and mitochondrial stress (*hsp-60*). In the *flp-7$^{tg}$* animals (*Figure 2C–E*) and in those treated with 5-HT (*Figure 2—figure supplement 1*), we observed a two-fold induction of an integrated *hsp-60::GFP* transgene, the canonical reporter for the mitochondrial stress response. Other stress responses were absent. The induced mitochondrial *hsp-60*-mediated stress response was seen predominantly in the intestine (*Figure 2C*) and was wholly dependent on the presence of *atgl-1* because RNAi-mediated inactivation of *atgl-1* ameliorated *hsp-60* induction as judged by the reporter assay (*Figure 2D*) as well as by directly measuring *hsp-60* transcripts by qPCR (*Figure 2E*). The increased mitochondrial respiration in *flp-7$^{tg}$* also depended on *atgl-1*-dependent utilization of fat reserves (*Figure 1H,I*), suggesting the surprising result that fat oxidation evokes a mitochondrial stress response.

### The mitochondrial stress response factor ATFS-1 sustains fat oxidation via ATGL-1

All known mitochondrial stress pathways that induce *hsp-60* require the mito-nuclear transcription factor called ATFS-1 (*Nargund et al., 2012*). Under normal conditions, ATFS-1 is transported into mitochondria and degraded. However, under conditions that induce mitochondrial stress, ATFS-1 is

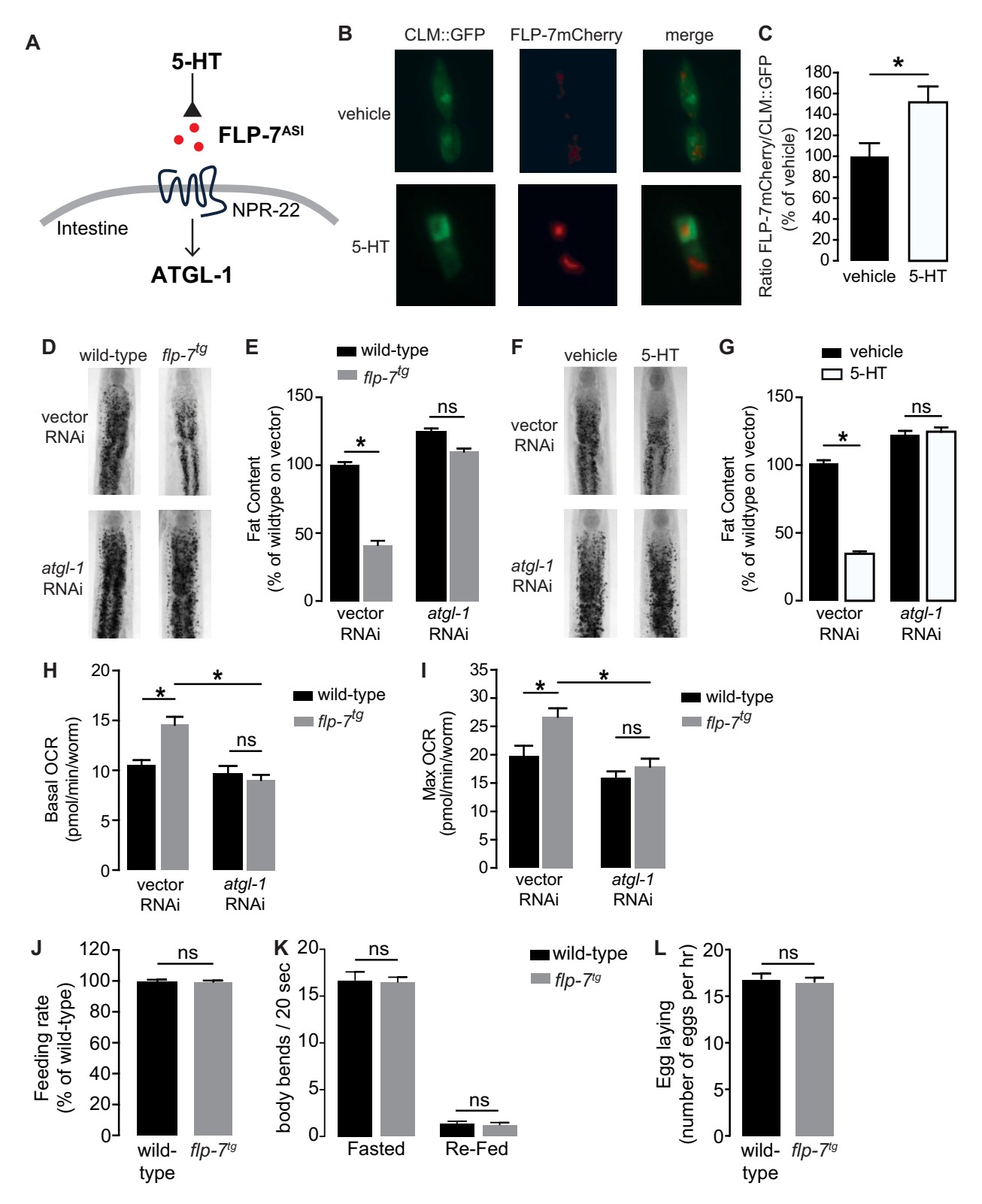

**Figure 1.** Neuronal FLP-7/tachykinin secretion as an experimental tool to model sustained fat oxidation in vivo. (**A**) Model depicting the *C. elegans* tachykinin neuroendocrine axis. Neuronal FLP-7/tachykinin is secreted from the ASI neurons in response to 5-HT signaling. Secreted FLP-7 acts on its cognate receptor NPR-22 (ortholog of the mammalian neurokinin two receptor NK2R) in the intestine. FLP-7/NPR-22 signaling transcriptionally activates the conserved lipase ATGL-1, which is the rate-limiting enzyme that converts stored triglycerides to free fatty acids destined for beta-oxidation. (**B**) An

*Figure 1 continued on next page*

*Figure 1 continued*

assay to detect and quantify secreted FLP-7mCherry that was previously validated (*Palamiuc et al., 2017*). Representative images of vehicle or 5-HT treated wild-type animals bearing a FLP-7mCherry fusion protein (red) expressed from ASI neurons, and coelomocytes countermarked with GFP (CLM:: GFP). FLP-7mCherry is secreted and non-specifically endocytosed by the coelomocytes. Left panels, GFP expression in coelomocytes; center panels, secreted FLP-7mCherry in coelomocytes; Right panels, merge. (C) The intensity of FLP-7mCherry within the coelomocytes was quantified and normalized to the area of CLM::GFP. Data are presented as percent of vehicle ± sem (n = 21). *, p<0.05 by Student's t-test. (D) Representative images of wild-type and *flp-7^tg* animals on vector or *atgl-1* RNAi fixed and stained with Oil Red O. (E) Quantification of lipid droplets in the depicted conditions, presented as a percentage of wild-type controls (n = 16–20). ns, not significant. *, p<0.05 by two-way ANOVA. (F) Representative images of wild-type animals on vector or *atgl-1* RNAi, treated with vehicle or 5-HT, fixed and stained with Oil Red O to measure fat content. (G) Quantification of lipid droplets in the depicted conditions, presented as percent of wild-types treated with vehicle ± sem. (n = 18–20). ns, not significant. *, p<0.05 by two-way ANOVA. (H, I) Oxygen consumption rate (OCR) of wild-type or *flp-7^tg* animals fed vector or atgl-1 RNAi. Basal OCR (H) was quantified prior to addition of FCCP (50 µM) and maximal OCR (I) was determined following FCCP stimulation. Data are presented as pmol/min/worm ± sem. (n = 10–20 wells, each containing approximately 10 worms). For all genotypes and conditions tested, no developmental delays, growth or size changes were detectable. ns, not significant. *, p<0.05 by two-way ANOVA. (J) Feeding rate was measured in wild-type and *flp-7^tg* animals and data are presented as percent of wild-type. (n = 25 per group) ns, not significant by Student's t-test. (K) Locomotion was measured in wild-type and *flp-7^tg* animals that were fasted or animals that were fasted and then re-fed. Data are plotted as body bends per 20 s. (n = 15 per group) ns, not significant by two-way ANOVA. (L) Egg laying was recorded in wild-type and *flp-7^tg* animals and results are presented as number of eggs laid per hour. (n = 10 per group) ns, not significant by Student's t-test.

stabilized and translocated to the nucleus where it initiates a range of transcriptional responses that allow adaptation to the stressor (*Lin and Haynes, 2016*; *Nargund et al., 2015*). To determine whether the FLP-7-induced mitochondrial stress response required *atfs-1*, we measured *hsp-60* induction in its absence. As shown in *Figure 2F and G*, *hsp-60* was no longer induced by FLP-7 in the *atfs-1(tm4525)* null mutants. We also noted that the combined loss of *atgl-1* and *atfs-1* completely suppressed the *hsp-60* induction, which recapitulates the loss of each gene alone with no additive or synergistic effects (*Figure 2F,G*). This finding was replicated with 5-HT-stimulated *hsp-60* induction that was completely suppressed with either *atgl-1* or *atfs-1* inactivation (*Figure 2— figure supplement 1*). This result suggested the possibility that *atgl-1*-dependent fat oxidation and *atfs-1*-dependent mitochondrial stress may function in a linear pathway.

Despite our knowledge about the role of ATFS-1 in mitochondrial biology, it had not previously been associated with fat oxidation per se. To our surprise, we found that 5-HT-induced fat loss was partially suppressed in *atfs-1* null mutants (*Figure 3—figure supplement 1A*), and FLP-7-induced fat loss was fully suppressed in *atfs-1* null mutants (*Figure 3A,B*). This discrepancy in partial versus complete suppression likely results from the widely-noted pleiotropic effects of exogenous 5-HT administration (*Palamiuc et al., 2017*; *Srinivasan et al., 2008*). The absence of *atfs-1* also suppressed the transcriptional induction of *atgl-1* by FLP-7 and 5-HT as judged by measuring the integrated *atgl-1* reporter in vivo (*Figure 3C,D*), and by direct measurement of *atgl-1* transcripts by qPCR (*Figure 3E* and *Figure 3—figure supplement 1B*). As in the case of the *hsp-60*-mediated stress response (*Figure 2D,E*), loss of both *atfs-1* and *atgl-1* also did not lead to a further suppression of FLP-7-induced fat loss (*Figure 3A,B*), suggesting again that they function in a linear pathway. However, these data also indicated that the effects of secreted FLP-7 on the intestine are intertwined: on the one hand the mitochondrial stress response requires *atgl-1* and fat oxidation (blue arrow; *Figure 3F*), and on the other hand fat oxidation requires the mito-nuclear stress response transcription factor *atfs-1* (green arrow; *Figure 3F*). Published ChIPseq and microarray studies of the ATFS-1 transcription factor did not suggest a direct induction of *atgl-1* by ATFS-1 (*Nargund et al., 2015*; *Nargund et al., 2012*). We also did not find the putative cis-binding site of ATFS-1 within a 5 kb region upstream of the *atgl-1* transcriptional start site. Yet, ATFS-1 is required for both the induction of *atgl-1* expression (*Figure 3C–E*), and the ensuing fat loss (*Figure 3A,B*). Thus, ATFS-1 likely regulates *atgl-1* transcription by an indirect mechanism.

## The conserved transcription factor HLH-11 governs fat oxidation via direct control of ATGL-1

To identify direct transcriptional regulators of *atgl-1* in vivo, we began by conducting an RNAi-based screen of the ~900 transcription factors in *C. elegans* (*Fuxman Bass et al., 2016*; *MacNeil et al., 2015*). We used the *Patgl-1::GFP* reporter we had previously developed (*Noble et al., 2013*) and screened for genes that regulate *atgl-1* under basal conditions, as well as for those essential for 5-

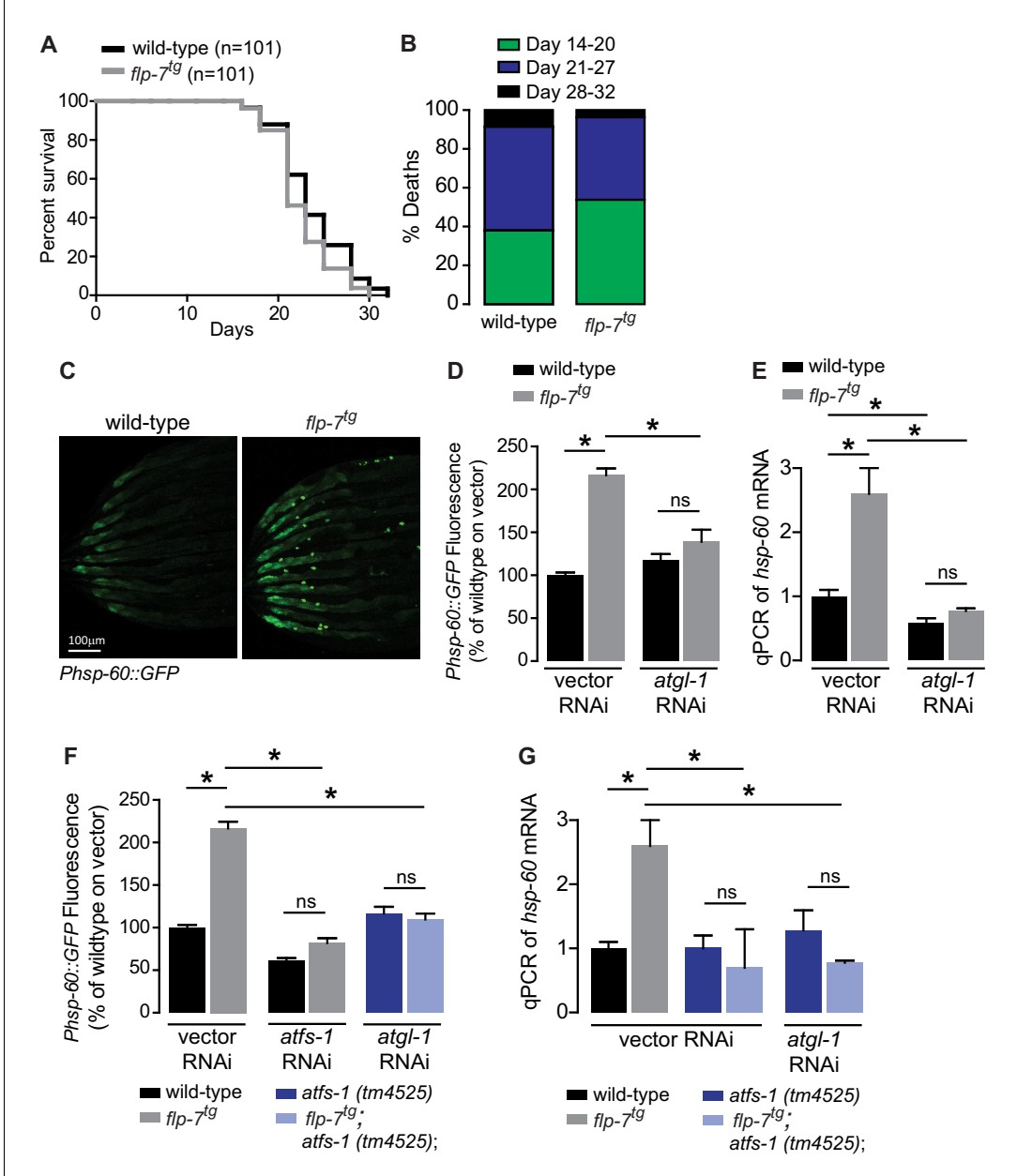

**Figure 2.** Sustained fat oxidation in the intestine stimulates a mitochondrial stress response. (A) Lifespans of wild-type (n = 101) and *flp-7*[tg] (n = 101) were assessed by counting the number of alive and dead animals every other day until all animals had died. Data are plotted as the percentage of animals that survived on any given day, relative to the number of animals alive on Day 1 of adulthood. p-value=0.08 by the Log-Rank Test. (B) Percent deaths of wild-type and *flp-7*[tg] animals over the span of days 14–20 (green), days 21–27 (blue), and days 28–32 (black). Data are presented as percent death for each range of days. (C) Representative images of wild-type and *flp-7*[tg] bearing an integrated *Phsp-60::GFP* transgene. Scale bar, 100 µm. (D) GFP intensity was quantified and normalized to the area of each animal, fed vector or *atgl-1* RNAi, expressed relative to wild-type vector ± sem. (n = 30). ns, not significant. *, p<0.05 by two-way ANOVA. (E) *hsp-60* mRNA was measured via qPCR in the groups indicated. *act-1* mRNA was used as a control. Data are presented as fold change relative to wild-type vector ± sem. (n = 4–6). ns, not significant. *, p<0.05 by two-way ANOVA. (F) The fluorescence intensity of *hsp-60* expression was quantified in the conditions indicated in the figure panel. Data are presented as a percent of wild-type vector ± sem. (n = 30). ns, not significant. *, p<0.05 by two-way ANOVA. (G) qPCR of *hsp-60* mRNA. Data are presented as fold change relative to wild-type vector ± sem. (n = 4–6). ns, not significant. *, p<0.05 by two-way ANOVA.

The online version of this article includes the following figure supplement(s) for figure 2:

**Figure supplement 1.** Induction of mitochondrial stress by 5-HT.

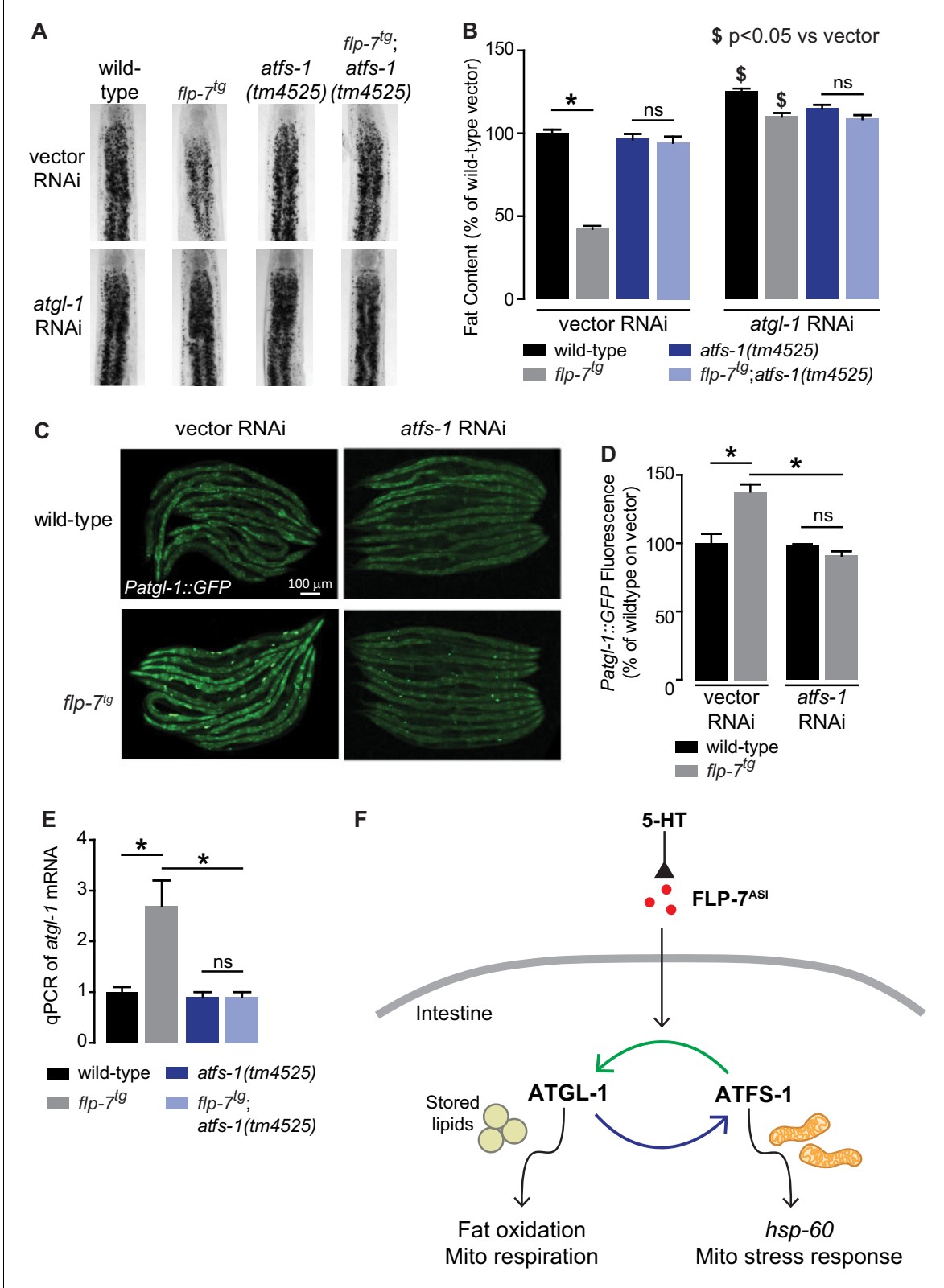

**Figure 3.** The ATFS-1-mediated mitochondrial stress response is required to sustain fat oxidation. (A) Representative images of wild-type and indicated genotypes on vector or *atgl-1* RNAi fixed and stained with Oil Red O. (B) Quantification of lipid droplets in the depicted conditions, presented as a percentage of wild-type vector ± sem. (n = 20). ns, not significant. *, p<0.05 vs wild-type and $ p<0.05 vs vector by two-way ANOVA. (C) Representative images are shown of wild-type and *flp-7^tg^* animals bearing an integrated *Patgl-1::GFP* transgene fed vector or *atfs-1* RNAi. Scale bar, 100 μm. (D) The
*Figure 3 continued on next page*

*Figure 3 continued*

fluorescence intensity of *atgl-1* expression is quantified and presented as percent of wild-type vector ± sem. (n = 29–30). ns, not significant. *, p<0.05 by two-way ANOVA. (**E**) qPCR of *atgl-1* mRNA. *act-1* mRNA was used as a control. Data are presented for the indicated genotypes as fold change relative to wild-type ± sem. (n = 4). ns, not significant. *, p<0.05 by two-way ANOVA. (**F**) As indicated by the data, model depicting the reciprocal regulatory relationship between the fat-burning enzyme ATGL-1 that triggers the *hsp-60* mitochondrial stress response (blue arrow), and the stress sensor ATFS-1 that is required for fat oxidation (green arrow).

The online version of this article includes the following figure supplement(s) for figure 3:

**Figure supplement 1.** 5-HT drives fat metabolism via ATFS-1.

HT or FLP-7-stimulated *atgl-1* induction. Our top hit was a gene called *hlh-11*, the sole conserved ortholog of the mammalian transcription factor AP4 (*Lee et al., 2009*). We obtained and outcrossed a null mutant for *hlh-11(ok2944)*, which when crossed into the *Patgl-1::GFP* reporter line showed significant in vivo induction of *atgl-1* in the intestine (*Figure 4A,B*). These results were reinforced by measuring transcript levels of *atgl-1* mRNA by qPCR, which were significantly greater in *hlh-11* null mutants (*Figure 4C*). We also noted that the extent of *atgl-1* induction by FLP-7 was matched by the absence of *hlh-11*, and without a further increase in *flp-7^{tg}*;*hlh-11* animals (*Figure 4A–C*). These data suggested that in wild-type animals, HLH-11 functions as a repressive transcription factor that suppresses *atgl-1* under basal conditions. In turn, this predicts that a metabolic phenotype may result from *hlh-11* removal. *hlh-11* null mutants showed a dramatic ~70–80% decrease in body fat stores relative to wild-type, and phenocopied the FLP-7 transgenic line (*Figure 4D,E*). *flp-7^{tg}*;*hlh-11* animals also had significant reductions in body fat stores and resembled either single mutant alone. Removal of *atgl-1* by RNAi completely suppressed fat loss under all conditions (*Figure 4D,E*). Further, the fat loss phenotype of *hlh-11* null mutants was accompanied by a significant increase in basal and maximal respiration that was also abrogated in the absence of *atgl-1* (*Figure 4F,G*). Together, these data show that loss of the transcription factor *hlh-11* constitutively increases *atgl-1* gene expression, fat oxidation, and energy expenditure.

To test whether HLH-11 was also instructive in regulating *atgl-1* expression and subsequent fat metabolism, we generated an integrated HLH-11 overexpression line (henceforth HLH-11^{ox}) fused to GFP using the endogenous 3 kb promoter (*Lee et al., 2009*). We observed robust nuclear HLH-11 expression in the intestine (*Figure 5A*), suggesting a plausible model for the interaction between HLH-11 and *atgl-1*. Remarkably, HLH-11^{ox} animals showed an ~40% increase in body fat stores compared to wild-type animals and showed a near-complete suppression of the fat loss elicited in *flp-7^{tg}* animals (*Figure 5B,C*). The increase in fat stores was accompanied by a corresponding decrease in maximal respiration, which was not further decreased upon removal of *atgl-1* (*Figure 5D*). The fat accumulation and metabolic output in HLH-11^{ox} and *flp-7^{tg}*;HLH-11^{ox} animals were accompanied by a significant decrease in *atgl-1* mRNA, as judged by qPCR (*Figure 5E*). Thus, HLH-11 plays an instructive role as a negative regulator of *atgl-1* expression and influences fat oxidation and mitochondrial respiration in the intestine.

The *cis*-binding site of HLH-11 has been precisely mapped to an 8-mer (*Grove et al., 2009*; *Lee et al., 2009*), but has not yet been functionally tested. We identified two HLH-11 cis-binding sites at 891 and 2,386 bp upstream of the *atgl-1* transcriptional start site (*Figure 5F*). To test for direct binding of the *atgl-1* promoter by HLH-11, we conducted chromatin immunoprecipitation experiments followed by qPCR (ChIP-qPCR) of the integrated transgenic line (*Phlh-11::hlh-11GFP*) in which HLH-11 was fused to GFP. Using three independent primer sets (*Figure 5F*), we observed *atgl-1* promoter regions bound to HLH-11GFP in vehicle-treated animals, suggesting that HLH-11 is constitutively bound to the *atgl-1* promoter under basal conditions (*Figure 5G*). Importantly, 5-HT treatment significantly reduced *atgl-1* promoter occupancy by HLH-11 relative to vehicle treated animals (*Figure 5G*). Thus, the transcription factor HLH-11 binds directly to the *atgl-1* promoter, and 5-HT signaling diminishes this interaction, ultimately promoting fat loss. Finally, we generated an extra-chromosomal *atgl-1* reporter line that lacked both *hlh-11* cis sites (Δcishlh-11) and observed an ~2-fold increase in *atgl-1* expression (*Figure 5H,I*). Taken together, the conserved transcription factor HLH-11 is a direct repressor of *atgl-1* expression, fat loss, and mitochondrial respiration.

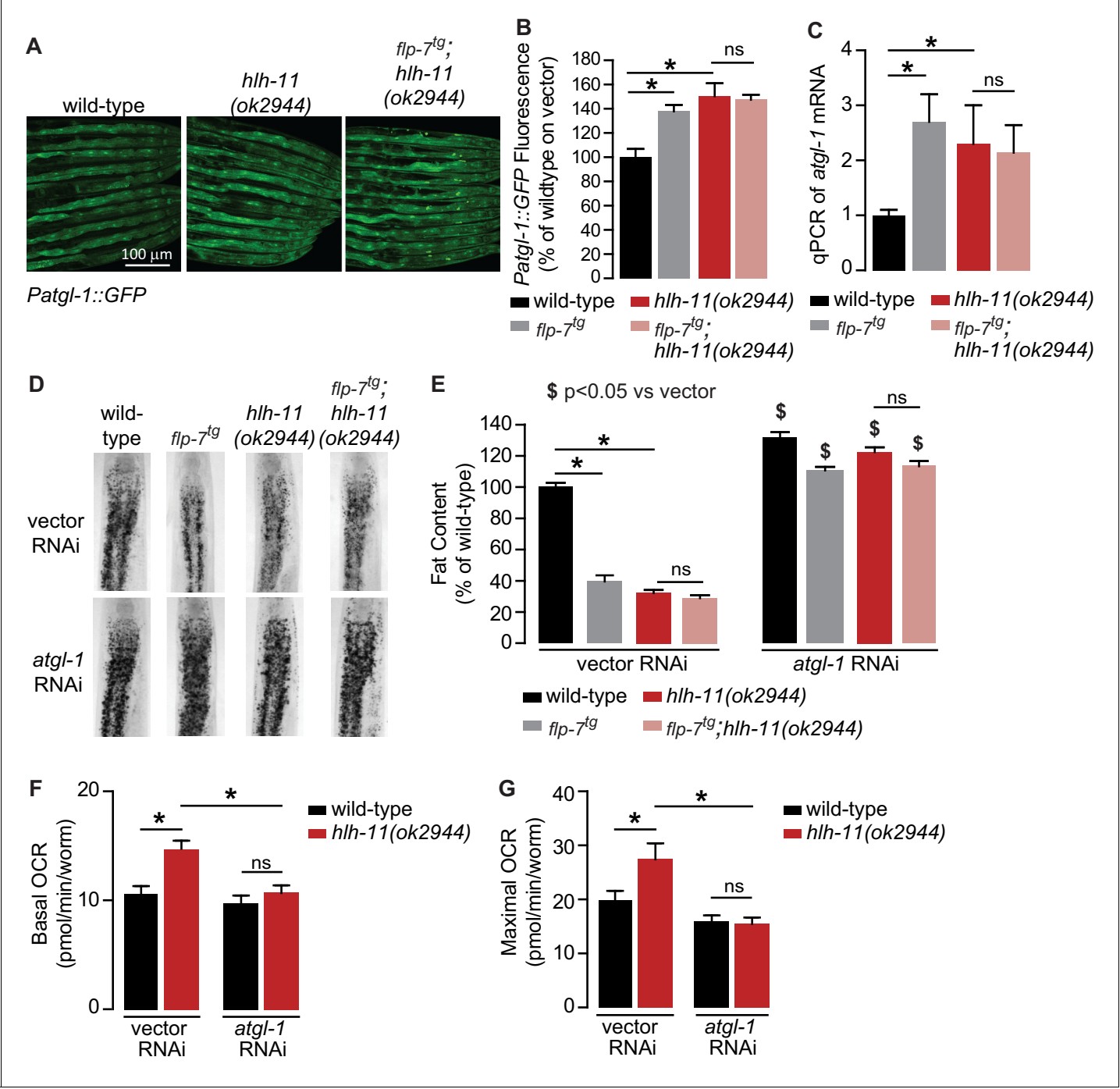

**Figure 4.** A genetic screen identifies the conserved transcription factor HLH-11 as a negative regulator of ATGL-1. (A) Representative images of wild-type, *flp-7*[tg], and *flp-7*[tg];*hlh-11(ok2944)* animals bearing an integrated *Patgl-1::GFP* transgene. Scale bar, 100 μm. (B) The fluorescence intensity of *atgl-1* expression was quantified and normalized to the area of each animal for each indicated genotype. Data are presented as percent of wild-type vector ± sem. (n = 26–30). ns, not significant. *, p<0.05 by one-way ANOVA. (C) qPCR of *atgl-1* mRNA in the indicated genotypes. *act-1* mRNA was used as a control. Data are presented as fold change relative to wild-type ± sem. (n = 4). ns, not significant. *, p<0.05 by one-way ANOVA. (D) Representative images of animals of the indicated genotypes on vector or *atgl-1* RNAi, fixed and stained with Oil Red O to measure fat content. (E) Quantification of lipid droplets in the depicted conditions, presented as percent of wild-types treated with vehicle ± sem. (n = 18–20). ns, not significant. *p<0.05 vs wild-type and $ p<0.05 vs vector by two-way ANOVA. (F, G) OCR of wild-type or *hlh-11(ok2944)* animals on vector or *atgl-1* RNAi were measured over time. Basal OCR (F) was quantified prior to addition of FCCP (50 μM) and maximal OCR (G) was determined following FCCP stimulation. Data are presented as pmol/min/worm ± sem. (n = 15 wells, each well containing approximately 10 worms). ns, not significant. *, p<0.05 by two-way ANOVA.

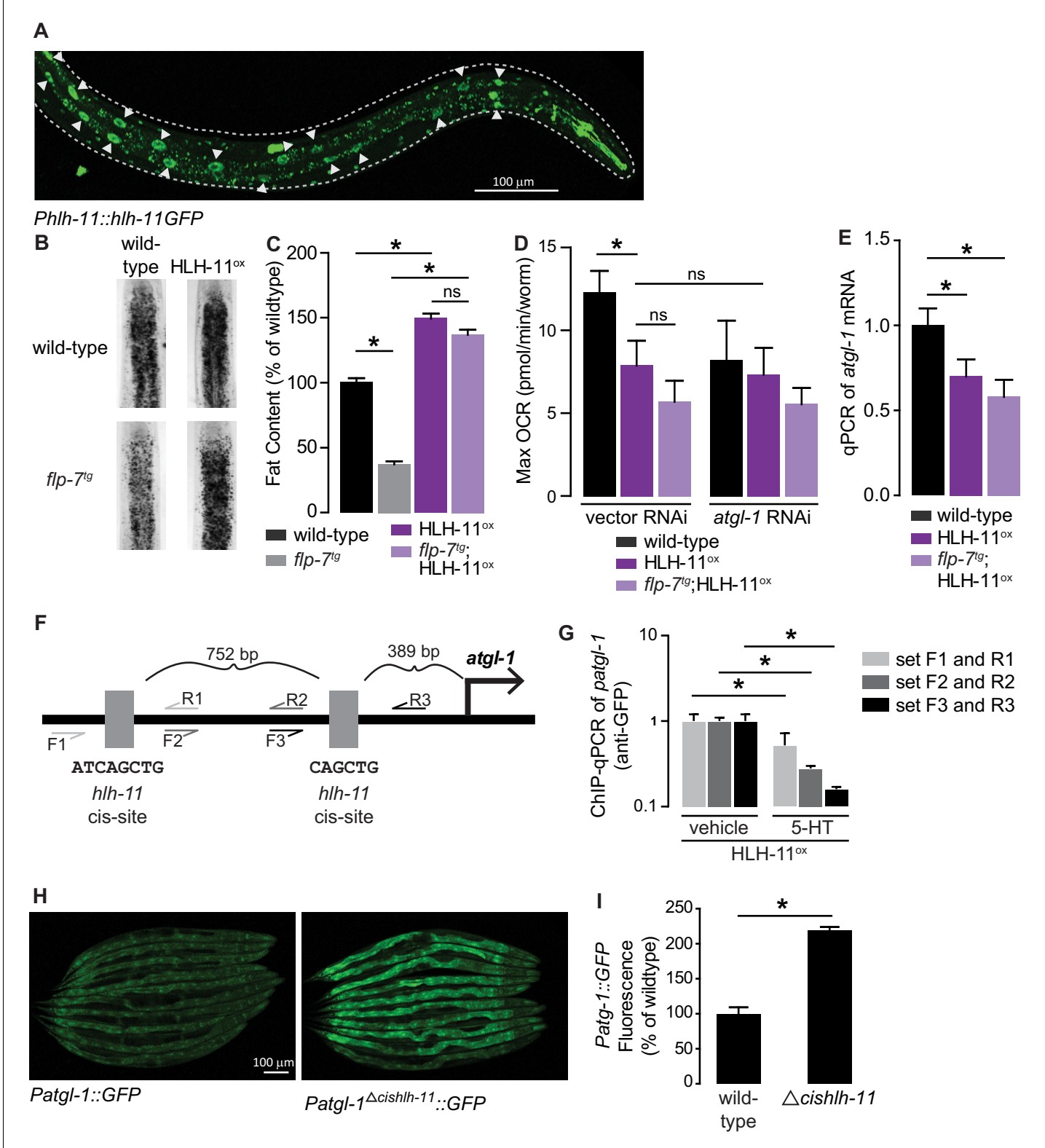

**Figure 5.** HLH-11 is a direct repressor of *atgl-1* transcription. (**A**) Representative image of an integrated transgenic animal expressing the HLH-11GFP fusion protein. GFP was clearly visible in the nucleus of intestinal cells (arrowheads). Scale bar, 100 μm. (**B**) Representative images of animals of the indicated genotypes, fixed and stained with Oil Red O to measure fat content. (**C**) Quantification of lipid droplets in the depicted conditions, presented as percent of wild-types ± sem. (n = 18–20). ns, not significant. (**D**) Maximal OCR of wild-type, HLH-11ᵒˣ, and *flp-7ᵗᵍ*;HLH-11ᵒˣ animals on vector or *atgl-1* RNAi. Data are presented as pmol/min/worm ± sem. (n = 15 wells each containing approximately 10 worms). ns, not significant. *, p<0.05 vs wild-type

*Figure 5 continued on next page*

Figure 5 continued

and $, p<0.05 vs vector RNAi by one-way ANOVA. (E) qPCR of *atgl-1* mRNA in wild-type, HLH-11^ox, and *flp-7^tg*;HLH-11^ox. *act-1* mRNA was used as a control. Data presented as fold change relative to wild-type ± sem. (n = 4–6). ns, not significant. *, p<0.05 by one-way ANOVA. (F) Schematic of the promoter region of *atgl-1*. There are two *hlh-11* cis-sites (grey): one is 389 bp upstream of the *atgl-1* transcription start site and the second binding site is another 752 bp upstream. The exact HLH-11 predicted binding sequence is listed for each site. For ChIP-qPCR, three primer sets were designed. Set #1 (light grey) flanks the distal HLH-11 binding site, set #2 (dark grey) targets the region between the binding sites, and set #3 (black) flanks the proximal binding site relative to the transcriptional start site. (G) Animals bearing the extrachromosomal *Phlh-11::hlh-11GFP* transgene were treated with vehicle or 5-HT and subjected to ChIP-qPCR. Data are presented as fold change relative to vehicle. ChIP-qPCR was performed using three technical replicates, and the experiment was repeated three times. *, p<0.05 by Student's t-test. (H) Representative images of transgenic animals bearing the *Patgl-1::GFP* (left panel) and those bearing the *Patgl-1^Δcishlh-11::GFP* transgene (right panel), which lacks both HLH-11 binding domains. Scale bar, 100 μm. (I) The fluorescence intensity of *atgl-1* expression was quantified, normalized to number of worms, and presented as percent of wild-type ± sem. (n = 23–26). *, p<0.05 by Student's t-test.

## The mito-nuclear transcription factor ATFS-1 promotes fat oxidation via HLH-11 regulation

Next, we wished to examine how *hlh-11* itself is regulated in the context of the FLP-7 pathway. Because *atgl-1* is induced by FLP-7 (*Figure 3C–E*) but repressed by HLH-11 (*Figure 4A–C*), we hypothesized that FLP-7 signaling itself might repress *hlh-11*. We found that relative to wild-type, the *flp-7^tg* line had reduced HLH-11GFP in the nuclei of the intestinal cells (*Figure 6A,B*), suggesting that *hlh-11* is downregulated by secreted FLP-7. This observation was corroborated by measuring *hlh-11* transcripts using qPCR (*Figure 6C*). These experiments establish that neuronally-secreted FLP-7 negatively regulates intestinal HLH-11 that directly binds to and represses ATGL-1 transcription.

However, we had originally identified HLH-11 in light of our data suggesting that the mito-nuclear transcription factor ATFS-1 was indirectly required for FLP-7-mediated *atgl-1* induction and fat oxidation in the intestine (*Figure 3*). In seeking a greater understanding of how ATFS-1 may be connected to HLH-11, we serendipitously found that a previous report had listed *hlh-11* as one of the direct targets of ATFS-1 (*Nargund et al., 2015*). To test the possibility of an interaction between ATFS-1 and *hlh-11* in the context of FLP-7-mediated fat oxidation and mitochondrial stress, we measured *hlh-11* transcripts in the absence of *atfs-1*. We found that *atfs-1* removal blunted FLP-7-mediated *hlh-11* repression (*Figure 6C*), suggesting that FLP-7-mediated *hlh-11* repression requires *atfs-1*. In *flp-7^tg* animals, inactivation of *atgl-1*, a condition that would simultaneously block fat oxidation and *atfs-1* activation, also blunted this *hlh-11* repression, suggesting a connection between fat oxidation, *atfs-1* induction, and *hlh-11* repression. Loss of both *atfs-1* and *atgl-1* diminished the FLP-7-dependent suppression of *hlh-11* to a similar extent as loss of either gene alone (*Figure 6C*). Thus, the repression of *hlh-11* by neuronal FLP-7 signaling occurs via ATFS-1, which is stabilized during fat-oxidation-induced mitochondrial stress (*Figure 3*). These results suggest the interesting possibility of a feedback loop (*Figure 6D*): repression of *hlh-11* by FLP-7 permits *atgl-1*-dependent fat oxidation, which induces *atfs-1* and in turn represses *hlh-11* to sustain and augment fat oxidation. Predictions from this model are two-fold. On the one hand, the ATFS-1/HLH-11 interaction should modulate the *hsp-60*-dependent mitochondrial stress response, and on the other, it should also regulate the *atgl-1* transcriptional response (*Figure 6D*).

## A feedback loop orchestrates the relationship between fat oxidation, mitochondrial stress and longevity

To functionally test the feedback loop model (*Figure 6D*), we studied the relationships between HLH-11 and ATFS-1 in the context of FLP-7 signaling. First, we predicted that HLH-11, a transcription factor that suppresses *atgl-1* and fat oxidation, would also regulate the mitochondrial stress response, in an *atgl-1*-dependent manner. Accordingly, we found that overexpression of HLH-11, which represses *atgl-1* mRNA (*Figure 5E*), showed a significant decrease in *hsp-60* mRNA (*Figure 7A*). Thus, HLH-11^ox decreases not only fat loss and mitochondrial respiration (*Figure 5B,C, D*), but also the ensuing stress response. In contrast, *hlh-11* null mutants, which constitutively stimulate fat oxidation and mitochondrial respiration by modulating *atgl-1* transcription (*Figure 4*),

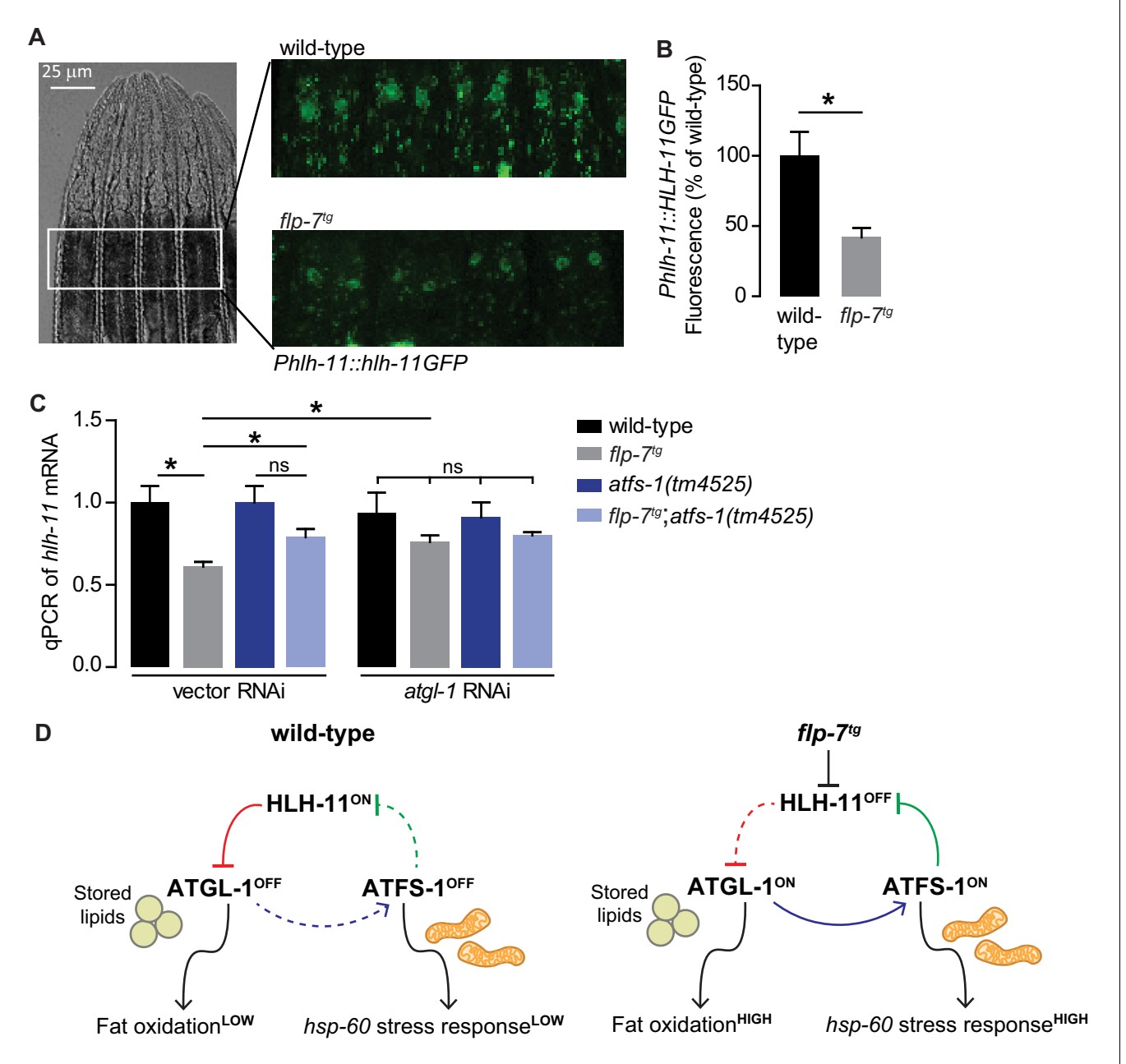

**Figure 6.** FLP-7 and ATFS-1 alter HLH-11 transcriptional status. (**A**) Representative images of wild-type and *flp-7^tg* animals bearing an integrated *Phlh-11::hlh-11GFP* transgene. DIC (left) and GFP (right) in the first two pairs of intestinal cells shown in representative animals. Scale bar, 25 µm. (**B**) The fluorescence intensity of *hlh-11* expression was quantified and normalized to the area of the first two pairs of intestinal cells. Data are presented as percent of wild-type vector ± sem. (n = 26–30). *, p<0.05 by Student's t-test. (**C**) qPCR of *hlh-11* mRNA in wildtype and *flp-7^tg* animals fed vector or *atgl-1* RNAi. *act-1* mRNA was used as a control. Data are presented as fold change relative to wild-type ± sem. (n = 4–6). ns, not significant. *, p<0.05 by two-way ANOVA. (**D**) Model for interaction between HLH-11, ATGL-1 and ATFS-1. In wild-type (left panel), HLH-11 is constitutively on, which represses ATGL-1 (red arrow) and keeps fat oxidation low. In this state, mitochondrial stress is not activated, *hsp-60* levels are low and ATFS-1 is not induced, keeping HLH-11 levels high. The FLP-7 neuronal signal (right panel) represses HLH-11, which de-represses ATGL-1, triggering fat oxidation. In turn, this generates an ATFS-1-dependent mitochondrial stress response as observed by the *hsp-60* induction (blue arrow). ATFS-1 represses HLH-11 (green arrow), providing a feedback cue to match fat oxidation with mitochondrial capacity.

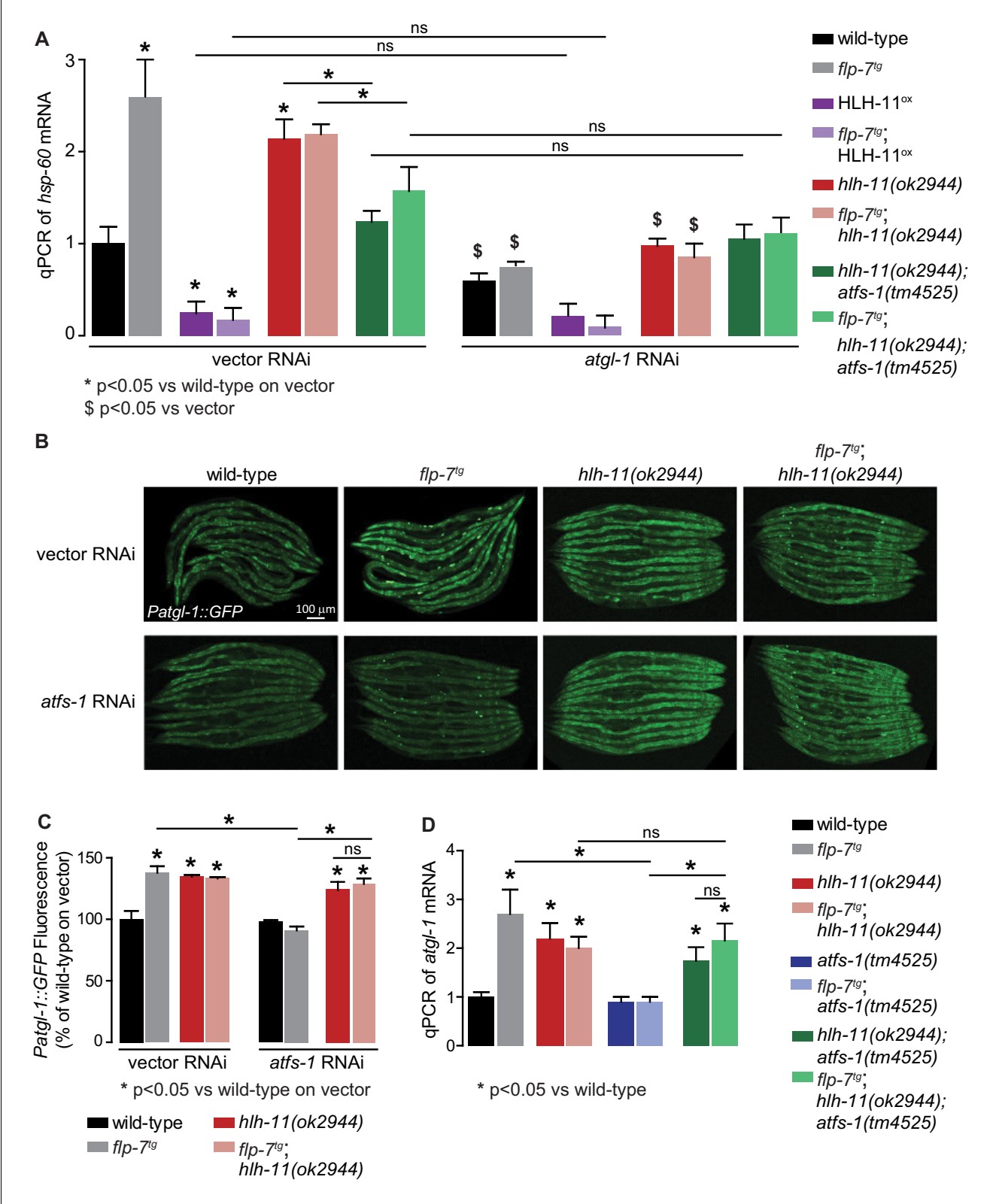

**Figure 7.** HLH-11-ATFS-1 interaction integrates HSP-60 mitochondrial stress response and ATGL-1 transcriptional status. (**A**) qPCR of *hsp-60* mRNA in groups indicated in the figure panel. Data are presented as fold change relative to wild-type vector ± sem. (n = 3–6). ns, not significant. *, p<0.05 vs wild-type on vector RNAi and $, p<0.05 vs vector RNAi by two-way ANOVA. (**B**) Representative images of animals bearing the integrated *Patgl-1::GFP* transgene. Animals with the indicated genotype were fed vector or *atfs-1* RNAi. Scale bar, 100 μm. (**C**) The fluorescence intensity of *atgl-1* expression

*Figure 7 continued on next page*

*Figure 7 continued*

was quantified and presented as percent of wild-type vector ± sem; groups as indicated in the figure panel. (n = 26–30). *p<0.05 vs wild-type on vector and $ p<0.05 vs vector by two-way ANOVA. (**D**) qPCR of *atgl-1* mRNA in the genotypes indicated on the right. Data presented as fold change relative to wild-type ± sem. (n = 3–6). *, p<0.05 vs wild-type by one-way ANOVA.

significantly increased *hsp-60* mRNA (***Figure 7A***). Effects of HLH-11 overexpression and absence were identical in the presence and absence of FLP-7 secretion (***Figure 7A***). Also, the induction of *hsp-60* upon loss of *hlh-11* was fully suppressed by removal of either *atgl-1*, *atfs-1*, or both, suggesting that HLH-11 mediated mitochondrial stress is a direct consequence of fat oxidation (***Figures 2F, G*** and ***7A***). Thus, rather than being independent of one another, HLH-11-mediated control of fat oxidation and energy expenditure evoke a stress response as a direct consequence of these mitochondrial functions.

Second, we tested the relationship between ATFS-1 and its transcriptional target *hlh-11*, with respect to *atgl-1* expression (refer to model in ***Figure 6D***). As expected, *hlh-11* mutants increased *atgl-1* expression with and without increased FLP-7 secretion as judged by *atgl-1* reporter expression (***Figure 7B,C***) as well as qPCR (***Figure 7D***). Also as predicted, *atfs-1* loss alone does not lead to appreciable changes in *atgl-1* (***Figure 7B–D***); this is because *atfs-1* is non-functional in wild-type animals (***Nargund et al., 2012***). In contrast, *hlh-11;atfs-1* double mutants resembled *hlh-11* mutants alone, thus during increased FLP-7 secretion, *atfs-1*-dependent induction of *atgl-1* requires *hlh-11* repression (***Figure 7B–D***). This result again suggested that rather than being a simple consequence of fat oxidation, the mitochondrial stress response is an integral component of sustained fat loss via *atgl-1* transcriptional regulation.

We were curious whether disrupting the HLH-11/ATGL-1/ATFS-1 feedback loop (***Figure 6D***) would shed light on the relationship between sustained fat oxidation and longevity. To this end, we measured the consequences of these transcriptional changes on physiological parameters. *flp-7^{tg}* animals have reduced fat stores because of increased *atgl-1*-dependent fat oxidation (***Figure 1***). In the context of *flp-7^{tg}* animals, *hlh-11* mutants also show augmented fat loss, whereas *atfs-1* removal blocked reduced fat stores (***Figure 8A,B***). In accordance with the increased *atgl-1* transcript levels (***Figure 7B–D***), we found that *hlh-11;atfs-1* double mutants also resembled *hlh-11* single mutants alone in their fat phenotypes, suggesting that the suppression of fat oxidation in *atfs-1* mutants is dependent on the presence of *hlh-11* (***Figure 8—figure supplement 1***). We note that in the presence of increased FLP-7 secretion, *hlh-11;atfs-1* double mutants showed a slight but significant augmentation of fat loss relative to *flp-7^{tg}* animals alone (***Figure 8A,B***); it is possible that of the hundreds of ATFS-1 targets, genes in addition to HLH-11 may play a role. Thus, ATFS-1-dependent fat oxidation requires *hlh-11* repression of *atgl-1* (***Figure 8A,B***); HLH-11 acts genetically downstream of ATFS-1.

We had already noted that increased fat loss via augmented FLP-7 secretion in *flp-7^{tg}* animals lead to no appreciable change in lifespan (***Figure 2A,B***). One explanation for this result is that FLP-7 signaling induces ATFS-1 activation because of mitochondrial fat oxidation. In addition to its role in repressing *hlh-11*, ATFS-1 has hundreds of additional targets which in combination have been postulated to serve mitochondrial recovery functions (***Lin and Haynes, 2016***; ***Nargund et al., 2012***). Thus, in the context of *flp-7^{tg}* animals, concomitant with the fat oxidation, our data suggest that an ATFS-1-dependent mechanism is simultaneously evoked, thus protecting lifespan. In testing this idea, we found that in *flp-7^{tg}* animals removal of both *hlh-11* and *atfs-1* led to a significant decrease in both median and maximal lifespan (p<0.001, ***Figure 8C,F***) that was not seen in *hlh-11;atfs-1* mutants (***Figure 8D***), *hlh-11* (***Figure 8E***) or *atfs-1* mutants (***Tian et al., 2016***). We reasoned that in *hlh-11;atfs-1* mutants alone (that is, without increased FLP-7 secretion) we observed modest fat loss (***Figure 8—figure supplement 1***) that is not sufficient to shift lifespan in either direction (***Figure 8D***). Even though *hlh-11* mutants show a substantial increase in fat oxidation (***Figure 4D,E***), the presence of *atfs-1* (***Figure 8E***) prevents lifespan shortening.

Why did loss of *hlh-11;atfs-1* in the *flp-7^{tg}* animals alone lead to a decrease in longevity? Our data and model (***Figure 8G***) are consistent with the following interpretation: FLP-7 secretion serves as the

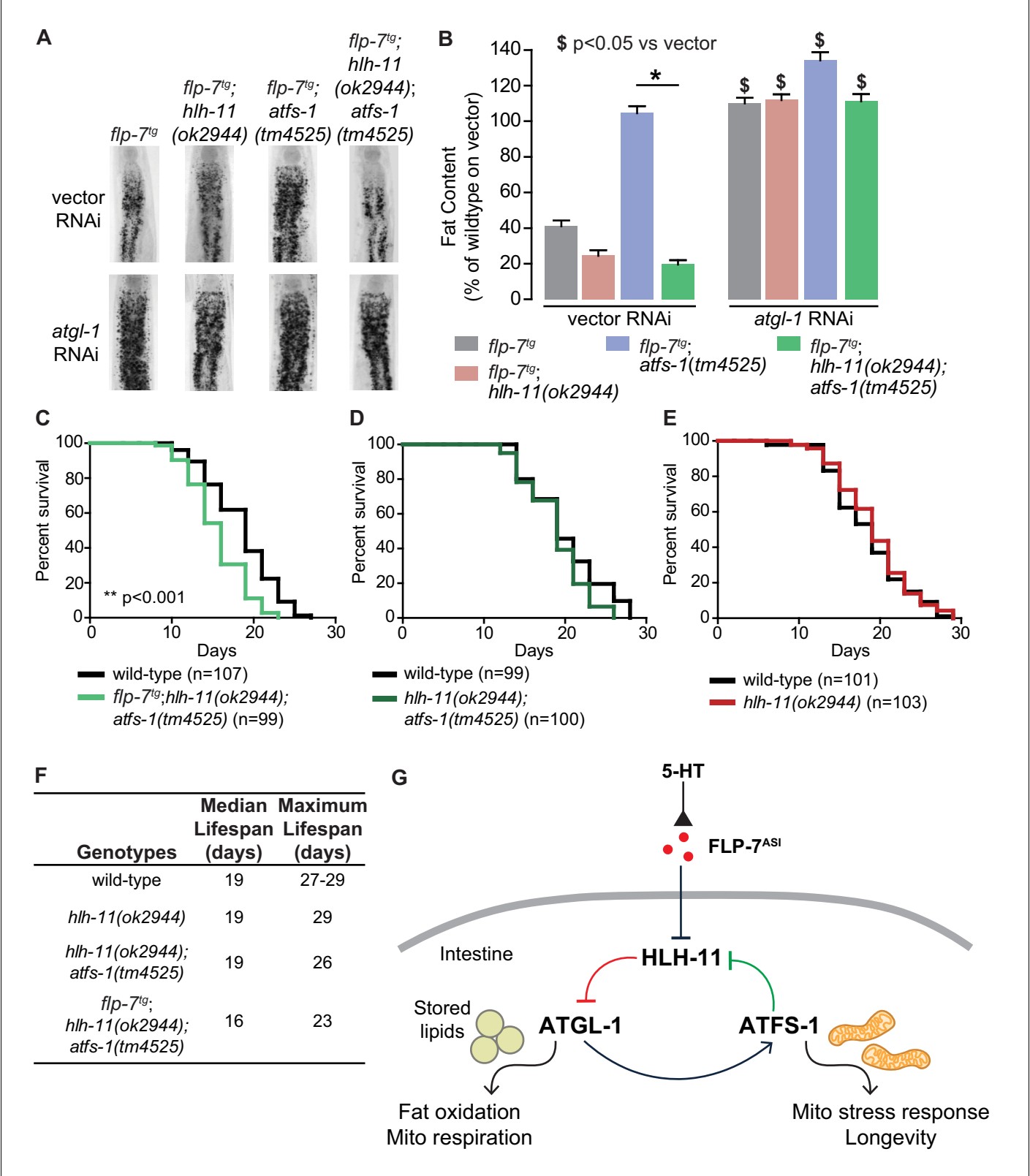

**Figure 8.** HLH-11 and ATFS-1 balance the tradeoff between sustained fat oxidation and longevity. (**A**) Representative images of animals of the indicated genotypes on vector or *atgl-1* RNAi, fixed and stained with Oil Red O to measure fat content. (**B**) Quantification of lipid droplets in the depicted conditions, presented as percent of wild-type vector ± sem. (n = 18–20). ns, not significant. *, p<0.05 vs wild-type and $, p<0.05 vs vector by

*Figure 8 continued*

two-way ANOVA. (**C–E**) Lifespan for each indicated genotype was assessed by counting the number of alive and dead animals every other day until all animals had died. Data are plotted as the percentage of animals that survived on any given day relative to the number of animals alive on Day 1 of adulthood. Number of animals used for each genotype are indicated in the figure panels. **p<0.001 by Log-Rank Test. (**F**) Table of median and maximum lifespans for each genotype. Data are presented in days. (**G**) Model depicting a feedback loop that coordinates fat oxidation with longevity. 5-HT-mediated FLP-7 secretion from neurons stimulates lipid metabolism in the intestine via repression of the conserved transcription factor HLH-11, which de-represses ATGL-1. Increased ATGL-1-dependent fat oxidation and mitochondrial respiration activates a mitochondrial stress response, as seen by induction of *hsp-60*. Simultaneously, the mito-nuclear transcription factor, ATFS-1, represses HLH-11 providing a feedback cue to sustain and augment HLH-11-dependent fat oxidation. On the one hand, overexpression of HLH-11 or absence of ATGL-1 blocks fat oxidation. On the other hand, during high fat oxidation and mitochondrial respiration, loss of the ATFS-1-HLH-11 feedback signal attenuates lifespan. Thus, the feedback signaling loop orchestrates the balance between sustained fat oxidation, mitochondrial stress, and lifespan.

The online version of this article includes the following figure supplement(s) for figure 8:

**Figure supplement 1.** Reduced fat content with loss of both *hlh-11* and *atfs-1*.

neuronal cue to trigger fat oxidation by repressing *hlh-11* expression, that in turn de-represses *atgl-1* (red pathway, *Figure 8G*). The resulting increase in fat oxidation and mitochondrial respiration generates an ATFS-1-mediated mitochondrial response, as judged by induction of the mitochondrial stress sensor *hsp-60*. We propose that ATFS-1 induction serves as a second cue from the mitochondria to further repress *hlh-11*, which in turn augments fat oxidation via ATGL-1 in a feedback loop (green pathway, *Figure 8G*). The *atgl-1/atfs-1/hlh-11* feedback loop serves the dual functions of sustaining fat oxidation and protecting lifespan. Although each influences the other, the *hlh-11/atgl-1* arm of the pathway primarily drives fat loss, whereas the *atfs-1/hlh-11* arm of the pathway primarily protects lifespan.

## Discussion

In this study, we have uncovered a regulatory pathway that is initiated in the nervous system and functions in the intestine, the major seat of metabolic and longevity regulation. The conserved transcription factor HLH-11 functions as the nexus of this signaling pathway and receives two signals: one non-cell-autonomous via neuronal FLP-7 that transmits sensory signals to initiate a metabolic response, and the other cell-autonomous via the mito-nuclear transcription factor, ATFS-1, that coordinates the mitochondrial stress response. HLH-11 is a direct transcriptional repressor of the triglyceride lipase *atgl-1*: loss of *hlh-11* increases *atgl-1*-dependent fat oxidation, and overexpression of *hlh-11* has the opposite effect. Thus, controlling *hlh-11* levels can serve as an excellent surrogate for titrating intestinal fat stores in future efforts. Interestingly, the mammalian ortholog of HLH-11, called AP4, is widely distributed along the epithelial cells of the mammalian intestine, as is ATGL (*Jung and Hermeking, 2009*; *Obrowsky et al., 2013*).

The broader goal of these studies was to address the question of whether, in a non-disease context, sustained fat oxidation and increased mitochondrial respiration have an effect on lifespan, because the relationship between fat oxidation and longevity, although of major significance, remained incompletely understood. We report here that sustained fat oxidation evokes a mitochondrial stress response that functions to simultaneously augment fat oxidation and confer longevity protection (modeled in *Figure 8G*). We have found that the dual control of fat oxidation and lifespan protection emerges from a feedback transcriptional loop that shields the organism from life-shortening mitochondrial stress in the face of continuous fat oxidation. Thus, sustained fat oxidation does not shorten lifespan because a mitochondrial response protects both. Under normal or wild-type conditions when fat oxidation levels are not high, this loop remains latent. We speculate that under conditions in which mitochondrial function is not compromised, the dissociation between adiposity and longevity may be broadly conserved.

# Materials and methods

## Key resources table

| Reagent type (species) or resource | Designation | Source or reference | Identifiers | Additional information |
|---|---|---|---|---|
| Strain, strain background (C. elegans) | N2;Pdaf-7::flp-mCherry; Punc-122::GFP | This paper | SSR1164 | Integrated. Backcrossed 6X. See Methods section "Cloning and transgenic strain construction" |
| Strain, strain background (C. elegans) | N2;Phsp-60::GFP | *Nargund et al., 2012* | | Integrated. |
| Strain, strain background (C. elegans) | N2;Pdaf-7::flp-mCherry; Punc-122:: GFP;Phsp-60::GFP | This paper | SSR1488 | Integrated. See Methods section "Cloning and transgenic strain construction" |
| Strain, strain background (C. elegans) | atfs-1(tm4525)V | *Nargund et al., 2012* | | |
| Strain, strain background (C. elegans) | atfs-1(tm4525); Pdaf-7::flp-7mCherry | This paper | SSR1481 | Integrated. See Methods section "Cloning and transgenic strain construction" |
| Strain, strain background (C. elegans) | atfs-1(tm4525); Phsp-60::GFP | *Nargund et al., 2012* | | Integrated. Backcrossed 1X. |
| Strain, strain background (C. elegans) | atfs-1(tm4525); Pdaf-7::flp-7mCherry; Punc-122::GFP; Phsp-60::GFP | This paper | SSR1571 | Integrated. Backcrossed 1X. See Methods section "Cloning and transgenic strain construction" |
| Strain, strain background (C. elegans) | N2;Patgl-1::GFP | This paper | SSR1564 | Integrated. Backcrossed 7X. See Methods section "Cloning and transgenic strain construction" |
| Strain, strain background (C. elegans) | N2;Pdaf-7::flp-7mCherry;Punc-122::GFP; Patgl-1::GFP | This paper | SSR1503 | Integrated. Backcrossed 7X. Originated from N2;Pdaf-7::flp-7mCherry; Punc-122::GFP and N2;Patgl-1::GFP |
| Strain, strain background (C. elegans) | hlh-11(ok2944)III | This paper | SSR725 | Backcrossed 7X. See Methods section "Worm maintenance and strains" |
| Strain, strain background (C. elegans) | hlh-11(ok2944); Pdaf-7::flp-7mCherry; Punc-122::GFP | This paper | SSR1560 | Integrated. Originated from hlh-11(ok2944) and N2;Pdaf-7::flp-7mCherry;Punc-122::GFP |
| Strain, strain background (C. elegans) | hlh-11(ok2944); Patg-1::GFP | This paper | SSR1563 | Integrated. Backcrossed 7X. Originated from hlh-11(ok2944) and N2;Patg-1::GFP |

*Continued on next page*

Continued

| Reagent type (species) or resource | Designation | Source or reference | Identifiers | Additional information |
|---|---|---|---|---|
| Strain, strain background (*C. elegans*) | *hlh-11(ok2944); Pdaf-7::flp-7mCherry;Punc-122::GFP; Patg-1l::GFP* | This paper | SSR1536 | Integrated. Backcrossed 7X. Originated from *hlh- 11(ok2944); Pdaf-7::flp-mCherry; Punc-122::GFP* and *N2;Patg-1::GFP* |
| Strain, strain background (*C. elegans*) | *N2;Phlh-11::hlh-11GFP* | This paper | SSR1530 | Integrated. Backcrossed 4X. See Methods section "Cloning and transgenic strain construction" |
| Strain, strain background (*C. elegans*) | *N2;Pdaf-7::flp-7mCherry;Punc-122::GFP;Phlh-11::hlh-11GFP* | This paper | SSR1550 | Integrated. Backcrossed 5X. Originated from *N2;Pdaf-7::flp-mCherry; Punc-122::GFP* and *N2; Phlh-11::hlh-11GFP* |
| Strain, strain background (*C. elegans*) | *N2;Patgl-1Δcishlh-11::GFP* | This paper | SSR1567 | Extrachromosomal. See Methods section "Cloning and transgenic strain construction" |
| Strain, strain background (*C. elegans*) | *hlh-11(ok2944); atfs-1(tm4525)* | This paper | SSR1399 | Originated from *hlh-11(ok2944)* and *atfs-1(tm4525)* |
| Strain, strain background (*C. elegans*) | *hlh-11(ok2944); atfs-1(tm4525); Pdaf-7::flp-7mCherry;Punc-122::GFP* | This paper | SSR1538 | Integrated. Originated from *hlh-11(ok2944);atfs-1 (tm4525)* and *N2; Pdaf-7::flp-7mCherry; Punc-122::GFP* |
| Antibody | GFP-Trap coupled to magnetic agarose beads (Camelidae antibody) | Bulldog Bio | Cat# GTMA020 | ChIP (20 uL per 3 mg of protein) |
| Recombinant DNA reagent | *Phlh-11::hlh-11GFP* (plasmid) | This paper | pSS1222 | See Methods section "Cloning and transgenic strain construction" |
| Recombinant DNA reagent | *Patgl-1::GFP* (plasmid) | *Noble et al., 2013* | pSS496 | |
| Recombinant DNA reagent | *Patgl-1Δcishlh-11::GFP* (plasmid) | This paper | pSS1245 | See Methods section "Cloning and transgenic strain construction" |
| Sequence-based reagent | Forward primer for cloning the promoter of *hlh-11* | IDT | | 5'-ggggacaactttgtat agaaaagttggagtg tggtgtgtttctcgtcag -3' |
| Sequence-based reagent | Reverse primer for cloning the promoter of *hlh-11* | IDT | | 5'-ggggactgctttttgta caaacttgtcattttcta ctattgatctacctg -3' |
| Sequence-based reagent | Forward primer for cloning the *hlh-11* gene | IDT | | 5'- ggggacaagtttgtac aaaaaagcaggcttgg ttcgttcggatag -3' |

*Continued on next page*

*Continued*

| Reagent type (species) or resource | Designation | Source or reference | Identifiers | Additional information |
|---|---|---|---|---|
| Sequence-based reagent | Reverse primer for cloning the *hlh-11* gene | IDT | | 5'- ggggaccactttgtaca agaaagctgggtaac ggacgagcgatgtctg -3' |
| Sequence-based reagent | Forward primer to remove the upstream *hlh-11 cis-site* in the *Patgl-1::GFP* plasmid | IDT | | 5'- gcactaactatttttt gttcgttcattttc -3' |
| Sequence-based reagent | Reverse primer to remove the upstream *hlh-11 cis-site* in the *Patgl-1::GFP* plasmid | IDT | | 5'- cctcctgtc tcggaacgc -3' |
| Sequence-based reagent | Forward primer to remove the downstream *hlh-11 cis-site* in the *Patgl-1::GFP* plasmid | IDT | | 5'- aatgattaca taaagtcacg -3' |
| Sequence-based reagent | Reverse primer to remove the downstream *hlh-11 cis-site* in the *Patgl-1::GFP* plasmid | IDT | | 5'- atcttgcta tgaatgtacc -3' |
| Sequence-based reagent | qPCR forward primer for *act-1* | IDT | | 5'- gtatggagtc cgccgga -3' |
| Sequence-based reagent | qPCR reverse primer for *act-1* | IDT | | 5'- cttcatggttga tggggcaa -3' |
| Sequence-based reagent | qPCR forward primer for *atgl-1* | IDT | | 5'- ctaccactgca atgggaatct -3' |
| Sequence-based reagent | qPCR reverse primer for *atgl-1* | IDT | | 5'- gtgggctgacc atatccaaata -3' |
| Sequence-based reagent | qPCR forward primer for *hlh-11* | IDT | | 5'- gcgcagaagaa gatcaaatcatc -3' |
| Sequence-based reagent | qPCR reverse primer for *hlh-11* | IDT | | 5'- ggtgccattc gtgcatttg -3' |
| Sequence-based reagent | qPCR forward primer for *hsp-60* | IDT | | 5'- cgagcttatcga gggaatgaa -3' |
| Sequence-based reagent | qPCR reverse primer for *hsp-60* | IDT | | 5'- gccttctcgta ctcgactttag -3' |
| Sequence-based reagent | ChIP-qPCR forward primer set 1 for *Patgl-1* | IDT | | 5'- cgtggggtac ggtacattca -3' |
| Sequence-based reagent | ChIP-qPCR reverse primer set 1 for *Patgl-1* | IDT | | 5'- ttggctagcg tgtagtgacg -3' |
| Sequence-based reagent | ChIP-qPCR forward primer set 2 for *Patgl-1* | IDT | | 5'- cgtcactacac gctagccaa -3' |
| Sequence-based reagent | ChIP-qPCR reverse primer set 2 for *Patgl-1* | IDT | | 5'- cagccaggtg gtgatggaat -3' |
| Sequence-based reagent | ChIP-qPCR forward primer set 3 for *Patgl-1* | IDT | | 5'- gtgatggcgt tccgagacag -3' |
| Sequence-based reagent | ChIP-qPCR reverse primer set 3 for *Patgl-1* | IDT | | 5'- ggtaccatac tggtacaaacg -3' |
| Chemical compound, drug | Serotonin (5-HT) | Alfa Aesar | Cat. # B21263-03 | 5 mM |
| Software, algorithm | ImageJ | *Schindelin et al., 2012* | | https://imagej.net/Fiji |
| Software, algorithm | GraphPad Prism | https://graphpad.com | | Version 8.0.0 |

## Materials availability

Further information and requests for resources and reagents should be directed to and will be fulfilled by the Lead Contact, Supriya Srinivasan (supriya@scripps.edu).

## Worm maintenance and strains

Nematodes were cultured as previously described (*Brenner, 1974*). N2 Bristol was obtained from the Caenorhabditis Genetic Center (CGC) and used as the wild-type reference strain. All mutant and transgenic strains used in the study are listed in the Key Resource Table. The *hlh-11(ok2944)* strain was provided by the *C. elegans* Gene Knockout Project at OMRF, which is part of the International *C. elegans* Gene Knockout Consortium. The *atfs-1(tm4525)* strain was generously provided by Cole Haynes (University of Massachussets Medical School, Worcester, MA) and originally obtained from the National BioResource Project (Tokyo, Japan). Animals were synchronized for experiments by hypochlorite treatment and then hatched L1 larvae were seeded on plates with the appropriate bacteria. All experiments were performed on synchronized day 1 adults, and there were no observable delays in growth, development or synchrony in the lines tested.

## Cloning and transgenic strain construction

The *flp-7$^{tg}$* line was previously developed (*Palamiuc et al., 2017*). Briefly, wild-type animals were injected with 5 ng/µL of the *Pdaf-7::flp-7mCherry* plasmid, 25 ng/µL of an *unc-122::GFP* plasmid, and 70 ng/µL of an empty vector. Approximately, 10–20 integrated transgenic lines were generated by UV crosslinking, and one line was selected for experimentation based on consistency of expression. This line was then backcrossed six times (SSR1164). SSR1164 was extensively validated to ensure that FLP-7 secretion in this line faithfully correlates with fluctuations in neuronal 5-HT signaling (*Palamiuc et al., 2017*). We also recapitulated the results from SSR1164 using an independent ASI-specific promoter *str-3* (SSR1297) to measure the fidelity of FLP-7 secretion (*Palamiuc et al., 2017*). The *hlh-11* promoter and gene were cloned using standard PCR techniques and Gateway Technology (Life Technologies) from N2 lysates. The final plasmid (*Phlh-11::hlh-11GFP*) encoded an HLH-11GFP fusion. 25 ng/µL of *Phlh-11::hlh-11GFP* plasmid was injected with 10 ng/µL of *Pmyo-2::mCherry* as a co-injection marker and 65 ng/µL of an empty vector to maintain a total injection mix concentration of 100 ng/µL. A well-transmitting transgenic line with consistent expression was integrated using the UV psoralen 2400 (Stratagene) and backcrossed four times before experimentation. To generate a plasmid of the *atgl-1* promoter lacking the two *hlh-11 cis-binding* sites (*Patgl-1$^{\Delta cishlh-11}$::GFP*), the Q5 site-directed mutagenesis kit (New England Biolabs) was used on the *Patgl-1::GFP* plasmid previously generated (*Noble et al., 2013*). The primers used are listed in the Key Resource Table. 25 ng/µL of the *Patgl-1$^{\Delta cishlh-11}$::GFP* plasmid, 50 ng/ µL of *rol-6* co-injection marker, and 25 ng/µL of empty vector were injected into wild-type worms. A transgenic line was selected based on consistency of expression and transmission.

## 5-HT treatment

5-HT hydrochloride powder (Alfa Aesar) was dissolved in 0.1 M HCl and was added to plates for a final concentration of 5 mM as previously described (*Palamiuc et al., 2017*).

## RNAi

RNAi experiments were conducted as previously described (*Kamath and Ahringer, 2003*; *Palamiuc et al., 2017*). Plates were seeded with HT115 bacteria containing vector or the relevant RNAi clone four days prior to seeding larvae.

## Oil Red O staining

Oil Red O staining was performed as described and validated (*Hussey et al., 2018*; *Hussey et al., 2017*; *Noble et al., 2013*; *Palamiuc et al., 2017*; *Witham et al., 2016*). Briefly, animals were washed off plates with PBS and incubated on ice for 10 min. Animals were fixed as described (*Noble et al., 2013*), after which they were stained in filtered oil Red O working solution (60% oil Red O in isopropanol: 40% water) overnight. For all genotypes and conditions, approximately 2000 animals were fixed and stained, and about 100 animals were visually examined. Then, 15–20 animals were chosen

blindly and imaged. All experiments were repeated at least three times. Wild-type animals were always included as controls within each experiment.

## Image acquisition and quantitation

Black and white images of oil Red O stained animals were captured using a 10X objective on a Zeiss Axio Imager microscope. Images were quantified using ImageJ software (NIH) as previously described (*Noble et al., 2013*). All reported results were consistent across biological replicates. Fluorescent images of reporters for FLP-7 secretion were captured using a 20X objective on a Zeiss Axio Imager microscope. The first pair of coelomocytes was imaged. mCherry fluorescence intensity in one of the two imaged coelomocytes was quantified and normalized to the area of the coelomocyte GFP as previously described and validated (*Palamiuc et al., 2017*). For fluorescence imaging of gene expression reporter lines (animals with integrated *Patgl-1::GFP*, *Phsp-60::GFP*, or *Phlh-11::hlh-11GFP* transgenes), an equal number of animals were chosen blindly and lined up side by side. Images were take using a 10X or 20X objective on a Nikon Eclipse 90i microscope. Fluorescence intensity for all chosen animals was quantified for each condition and normalized to area of the animals excluding the head as indicated in the figure legend. Images were quantified using ImageJ software (NIH).

## Oxygen consumption

Oxygen consumption rates (OCR) was measured using the Seahorse XFe96 Analyzer (Agilent) as previously described (*Hussey et al., 2018*). Briefly, adult animals were washed with M9 buffer and approximately 10 animals per well were placed into a 96-well plate. five measurements were taken for baseline, then at 37 min FCCP (50 μM) was injected to measure maximal OCR. Lastly, sodium azide (40 mM) was injected at 62 min to measure residual OCR. Afterwards, the number of worms per well was counted, and OCR values were normalized to number of worms per well. Basal OCR was calculated by averaging all measurements prior to FCCP (50 μM) addition, and maximal OCR was calculated by averaging the first two measurements after FCCP injection. In all genotypes tested, we did not observe any changes in worm size, growth or developmental stage.

## Behavioral assays

Feeding rate, locomotor response to food and egg-laying assays were conducted as described (*Palamiuc et al., 2017*). Feeding rate was measured by counting the contractions of the pharyngeal bulb over 10 s. For each genotype, 25 animals were counted and the experiment was repeated three times. To determine locomotor response, Day 1 adult animals were washed off plates with bacterial food. After five washes, animals were placed on NGM plates without bacteria for a 30 min fast. For the re-fed groups, animals were moved to food following the 30 min fast, allowed to acclimatize for 5 min following which the number of body bends per 20 s was counted. The egg-laying assay was conducted by moving five Day 1 adult animals to 3.5 cm plates seeded with OP50, and a total of 10 plates were used per group. After 3 hr, animals were removed from the plate and the number of eggs present on the plates were counted.

## Chromatin immunoprecipitation-qPCR

Chromatin immunoprecipitation was done as previously described (*Mukhopadhyay et al., 2008*). 50,000 synchronized D1 adult animals were washed with PBS three times and fixed with 1.1% formaldehyde for 15 min. Animals were partially lysed using a Dounce homogenizer. Fixative was quenched with 2.5 M glycine for 20 min. Animals were then washed and incubated in HEPES lyses buffer (50 mM HEPES-KOH pH 7.5, 150 mM NaCl, 1 mM EDTA, 0.1% (wt/vol) sodium deoxycholate, 1% (vol/vol) Triton X-100, 0.1% (wt/vol) SDS, 1 mM PMSF, and diluted protease inhibitor cocktail). Animal suspensions were sonicated using the Sonic Dismembrator Model 100 (Fischer Scientific) for 10 s and then place on ice for 2 min, which was repeated eight times. An aliquot of lysate was kept for input DNA analysis. 3 mg of protein was subjected to a pre-clearing step by incubating samples with prewashed salmon sperm DNA/protein-A agarose beads (Millipore Sigma) for 1 hr. The supernatant was then incubated with GFP-Trap coupled to magnetic agarose beads (Bulldog Bio) overnight at 4°C. As a negative control, wild-type animals, which lack GFP expression, were used. Bead-GFP-Trap-DNA complex was washed three times. Then precipitated DNA was eluted from the

beads, and the cross-link was reverse overnight at 65°C. Precipitated chromatin and the input samples were treated with proteinase K, and the DNA was purified using the PCR Purification kit (Qiagen). Purified DNA was then subjected to quantitative PCR using three sets of primers targeting the promoter region of *atgl-1*. Three technical replicates were used per group, and the entire ChIP-qPCR procedure was performed three times for a total of three biological replicates. Wild-type animals (without GFP), and primers for the *act-1* gene were used as negative controls. In the negative controls, any DNA detected in the qPCR (CT values were 32–38) was classified as background. ChIP samples were normalized to the corresponding input samples, and the fold change was calculated relative to the vehicle-treated group. All primer sequences are provided in the Key Resource Table.

## qPCR

Total RNA was isolated from D1 adult animals using TRIzol (Invitrogen) and purified as previously described (*Palamiuc et al., 2017*). cDNA was made using iScript Reverse Transcription Supermix for RT-qPCR kit (BioRad) following the manufacturer's instructions. SsoAdvanced Universal SYBR Green Supermix (BioRad) was used for performing qPCR according to the manufacturer's instructions. Data was normalized to *act-1* mRNA. Primers sequences are listed in the Key Resource Table. Fold change was calculated following the Livak method (*Livak and Schmittgen, 2001*). Each group had at least three biological replicates in which worms were harvested and RNA was isolated from each biological replicate separately. For each qPCR plate, at least two technical replicates were included per sample.

## Lifespans

All lifespan experiments were performed at 20°C (*Keith et al., 2014*). Approximately one hundred L4 larvae per group were transferred to NGM plates seeded with OP50, which was recorded as day 0. Animals were transferred to new plates every other day until egg laying stopped. Surviving and dead animals were counted every other day until all animals were dead. Animals were considered dead when they did not respond to a gentle stimulation with a platinum wire. Bagging, exploding, and contaminated animals were excluded from analysis.

## Statistical analyses

Each assay was powered for sample size and statistical test based on the following: (i) pilot studies to assess the strength of the phenotype; (ii) minimum number of animals needed to detect significant differences ($p<0.05$). For each type of assay, we used the same number of animals for each genotype or condition so as not to over-power or under-power comparisons. The sample size, statistical method and significance for each experiment is listed in the corresponding figure legend. All actual p-values are given in *Supplementary file 1*. Wild-type animals were included as controls for every experiment. Error bars represent standard error of the mean (sem). Student's t-test, one-way ANOVA, Log-Rank Test, and two-way ANOVA were used as indicated in the figure legends. Bonferroni's correction for multiple comparisons was used for all ANOVAs.

## Acknowledgements

Strains were provided by Knockout Consortium at Tokyo Women's Medical University as well as the CGC, which is funded by the NIH Office of Research Infrastructure Programs (P40 OD010440). This work was supported by research grants to SS from the NIH/NIDDK (R01 DK095804) and NIH/NIA (AG056648). NKL was supported by a fellowship from the American Heart Association (17POST33660740) and a Dorris Scholar Award from the Dorris Neuroscience Center, The Scripps Research Institute.

## Additional information

### Funding

| Funder | Grant reference number | Author |
| --- | --- | --- |
| National Institute of Diabetes and Digestive and Kidney Dis- | DK095804 | Supriya Srinivasan |

eases

| | | |
|---|---|---|
| National Institute on Aging | AG056648 | Supriya Srinivasan |
| American Heart Association | 17POST33660740 | Nicole K Littlejohn |
| Dorris Neuroscience Center (DNC) | Dorris Scholar Award | Nicole K Littlejohn |

The funders had no role in study design, data collection and interpretation, or the decision to submit the work for publication.

### Author contributions

Nicole K Littlejohn, Conceptualization, Resources, Data curation, Formal analysis, Funding acquisition, Investigation, Methodology, Writing - review and editing; Nicolas Seban, Conceptualization, Resources, Data curation, Formal analysis, Investigation, Methodology; Chung-Chih Liu, Formal analysis, Investigation, Methodology; Supriya Srinivasan, Conceptualization, Resources, Supervision, Funding acquisition, Methodology, Writing - original draft, Project administration, Writing - review and editing

### Author ORCIDs

Nicole K Littlejohn (iD) https://orcid.org/0000-0002-8755-967X
Supriya Srinivasan (iD) https://orcid.org/0000-0003-2544-3652

### Decision letter and Author response

Decision letter https://doi.org/10.7554/eLife.58815.sa1
Author response https://doi.org/10.7554/eLife.58815.sa2

## Additional files

### Supplementary files

• Supplementary file 1. List of all p-values and 95% Confidence Interval (CI) of the difference between means for all Figures. * indicates the 95% CI for the ratio of median survival.

• Transparent reporting form

### Data availability

All data generated or analysed during this study are included in the manuscript and supporting files.

The following previously published dataset was used:

| Author(s) | Year | Dataset title | Dataset URL | Database and Identifier |
|---|---|---|---|---|
| Nargund AM, Fiorese CJ, Pellegrino MW, Deng P, Haynes CM | 2015 | To identify ATFS-1 target genes during mitochondrial stress | https://www.ncbi.nlm.nih.gov/geo/query/acc.cgi?acc=GSE63803 | NCBI Gene Expression Omnibus, GSE63803 |

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
