## [Decision Letter]

**Acceptance summary:**

The manuscript is well written and carefully designed that provides novel insight into the non-linear relationship between fat metabolism, mitochondrial respiration and longevity.

**Decision letter after peer review:**

Thank you for submitting your article "A Feedforward Loop Governs The Relationship Between Lipid Metabolism And Longevity" for consideration by *eLife*. Your article has been reviewed by three peer reviewers, one of whom is a member of our Board of Reviewing Editors, and the evaluation has been overseen by David Ron as the Senior Editor. The following individual involved in review of your submission has agreed to reveal their identity: Albertha JM Walhout (Reviewer #3).

The reviewers have discussed the reviews with one another and I drafted this decision to help you prepare a revised submission.

We would like to draw your attention to changes in our revision policy that we have made in response to COVID-19 (https://elifesciences.org/articles/57162). Specifically, when editors judge that a submitted work as a whole belongs in *eLife* but that some conclusions require a modest amount of additional new data, as they do with your paper, we are asking that the manuscript be revised to either limit claims to those supported by data in hand, or to explicitly state that the relevant conclusions require additional supporting data. Our expectation is that the authors will eventually carry out the additional experiments and report on how they affect the relevant conclusions either in a preprint on bioRxiv or medRxiv, or if appropriate, as a Research Advance in *eLife*, either of which would be linked to the original paper.

Summary:

Littlejohn et al. discovers a mechanism that connects long term fat oxidation and mitochondrial stress response with lifespan. This is a timely study providing a new level of knowledge on the relationship between fat oxidation and longevity. The authors had previously discovered FLP-7/NPR-22 signalling as the neuroendocrine axis that drives 5-HT-induced intestinal fat loss in *C. elegans* [Palamiuc et al., 2017]. In the present study, the authors generate a transgenic line that allows selective manipulation of the FLP-7/NPR-22 neuroendocrine axis, independently of 5-HT, to test the consequences of neuronally driven sustained fat loss on longevity in a non-diseased state. The study reports a feedback (rather than feedforward) transcriptional loop which is triggered by ATGL-1 and involves the mito-nuclear stress response transcription factor ATFS-1, and a novel and highly conserved repressor of ATGL-1 named HLH-11/AP4. The authors conclude that such loop orchestrates the balance between sustained fat loss and longevity by simultaneously augmenting HLH-11/ATGL-1 dependent fat loss and mitochondrial respiration, as well as protecting life span through ATFS-1/HLH-11 dependent pathway. In general, the manuscript is carefully designed and provides novel insight into the non-linear relationship between fat metabolism, mitochondrial respiration and longevity. There is consensus among the reviewers that the concept of a feed-forward mechanism is not well substantiated by the data and confuses rather than clarifies the underlying biology. Hence we recommend that the title be edited and that the manuscript is extensively rewritten as per our suggestions below, such as to meet the criteria for publication at *eLife*.

1) It appears that the authors uncovered a feedback loop, rather than a feedforward loop. 5-HT represses HLH-11, which represses lipase – thus 5-HT activates lipase and β-oxidation. This in turn activates ATFS-1, leading to a stress response (which is protective for lifespan). ATFS-1 in turn feeds back to repress HLH-11, thereby sustaining β-oxidation. The authors should rewrite the appropriate sections of the manuscript and change the title accordingly.

2) "Thus, the *flp-7^tg^* line fully recapitulates the metabolic effects of genetic and pharmacological manipulation of neuronal 5-HT signalling without altering other 5-HT-mediated behaviors (Horvitz et al., 1982; Loer and Kenyon, 1993; Palamiuc et al., 2017; Song and Avery, 2012; Sze et al., 2000; Waggoner et al., 1998)…" The referenced articles do not provide any data to substantiate the claim cited above. Palamiuc et al., 2017 reported that the lack of *flp-7* gene did not change feeding behaviour, locomotor activity and reproduction. However, the data on the effects of ASI neuron-specific *flp-7* overexpression on the above-mentioned phenotypes were missing in the study of Palamiuc et al. and the authors do not show any data on 5-HT signalling regulating other phenotypes in *flp-7^tg^* line apart from fat loss. The authors should provide appropriate citations and specify the phenotypes, or tone down statements and complete the phenotyping with follow up studies. The authors show that both 5-HT and FLP-7 induced fat loss was partially repressed in *atfs-1* null mutants as shown in Figure 3A-B and Figure 3—figure supplement 1A. However, the data indicates a marked difference in fat loss following FLP-7 and 5-HT stimulation in this line. While the suppression of fat loss was partial in the *atfs-1* null mutants treated with 5-HT (Figure 3—figure supplement 1A),the FlP-7 induced fat loss was completely recovered in the *atfs-1* null mutants (Figure 3B). The authors could provide possible explanations for such discrepancy.

3) In Figure 6C, the data from *flp-7^tg^* and *flp-7^tg^*;*atfs-1* are missing in both vector and *atgl-1* RNAi conditions making difficult the interpretation of these results ¬- if the authors do not have these controls readily available then they should state this caveat, and provide the data in a follow up publication in bioRxiv or similar.

4) "In contrast, *hlh-11;atfs-1* double mutants resembled *hlh-11* mutants alone, thus during increased FLP-7 secretion, *atfs-1*-dependent suppression of *atgl-1* induction requires *hlh-11* repression".

a) In *flp-7^tg^* line, *atgl-1* expression significantly reduced in the absence of *atfs-1*^-1^(Figures 6F, 3D-E) indicating that atfs1 activates *atgl-1* during increased FLP-7 secretion. Also, *atfs-1* mediated suppression of *hlh-11* would induce *atgl-1* expression (Figure 6D). Therefore, the above statement does not match the observations and should be rephrased.

b) In general, the comprehensibility of Figure 6F would be greatly increased by adding representative fluorescent images as has been done with the preceding figures if the authors do not have these images readily available then they should state this caveat, and provide the data in a follow up publication in bioRxiv or similar.

5) Figure 7A and B show that during increased FLP-7 secretion, in the absence of both *hlh-11* and *atfs-1*, *flp-7^tg^* line lost almost half of their fat content although both lines have comparable level of *atgl-1* expression (Figure 6G, *flp-7^tg^* vs. *flp-7^tg^*;*hlh-11;atfs-1*). The authors could provide an explanation for these observations. Perhaps other signaling molecules might also be involved for fat loss in this model?

6) The authors normalize oxygen consumption in Figure 1H to worm rather than a measure of biomass such as protein. Does this mean that the worms are all the same size and all have the same biomass?

7) Do the *flp-7^tg^*, 5-HT treated animals, *hlh-11* mutant or the *hlh-11*^ox^ line show any alterations in fatty acid synthesis in parallel to the changes in OCR shown in Figures 1H, 4F and 5D?

8) From the model presented in Figure 3F, the authors imply that the induction of *atgl-1* is dependent upon the stress responses activated by *atfs-1* in an indirect manner. In line with this, they showed *atfs-1* induced repression of HLH-11 leads to *atgl-1* activation (Figure 5). However, the authors did not provide any data that could connect *hsp-60* stress response to *atgl-1* induction ¬- if the authors do not have this readily available then they should state this caveat, and provide the data in a follow up publication in bioRxiv or similar.

9) Age is a massive effector of fat mass accumulation. Are the mutants *flp-7^tg^*, *hlh-11*, and *atfs-1* or any combination therein developmentally altered to confound the interpretation of fat mass in Figure 7A-B?

Revisions expected in follow-up work:

As stated in points #3, 4b, and 8 above.

---

## [Author Response]

Summary:Littlejohn et al. discovers a mechanism that connects long term fat oxidation and mitochondrial stress response with lifespan. This is a timely study providing a new level of knowledge on the relationship between fat oxidation and longevity. The authors had previously discovered FLP-7/NPR-22 signalling as the neuroendocrine axis that drives 5-HT-induced intestinal fat loss in *C. elegans* [Palamiuc et al., 2017]. In the present study, the authors generate a transgenic line that allows selective manipulation of the FLP-7/NPR-22 neuroendocrine axis, independently of 5-HT, to test the consequences of neuronally driven sustained fat loss on longevity in a non-diseased state. The study reports a feedback (rather than feedforward) transcriptional loop which is triggered by ATGL-1 and involves the mito-nuclear stress response transcription factor ATFS-1, and a novel and highly conserved repressor of ATGL-1 named HLH-11/AP4. The authors conclude that such loop orchestrates the balance between sustained fat loss and longevity by simultaneously augmenting HLH-11/ATGL-1 dependent fat loss and mitochondrial respiration, as well as protecting life span through ATFS-1/HLH-11 dependent pathway. In general, the manuscript is carefully designed and provides novel insight into the non-linear relationship between fat metabolism, mitochondrial respiration and longevity. There is consensus among the reviewers that the concept of a feed-forward mechanism is not well substantiated by the data and confuses rather than clarifies the underlying biology. Hence we recommend that the title be edited and that the manuscript is extensively rewritten as per our suggestions below, such as to meet the criteria for publication at eLife.Revisions for this paper:1) It appears that the authors uncovered a feedback loop, rather than a feedforward loop. 5-HT represses HLH-11, which represses lipase – thus 5-HT activates lipase and β-oxidation. This in turn activates ATFS-1, leading to a stress response (which is protective for lifespan). ATFS-1 in turn feeds back to repress HLH-11, thereby sustaining β-oxidation. The authors should rewrite the appropriate sections of the manuscript and change the title accordingly.

Our claim of a feedforward loop was based on the phenotypic outcome of the regulatory loop, which is to sustain or augment fat loss initiated by the nervous system. However, at the genetic or pathway level, we agree that the mechanism occurs via feedback regulation. Accordingly, we have changed the title and made revisions throughout the text to reflect this.

2) "Thus, the flp-7^tg^ line fully recapitulates the metabolic effects of genetic and pharmacological manipulation of neuronal 5-HT signalling without altering other 5-HT-mediated behaviors (Horvitz et al., 1982; Loer and Kenyon, 1993; Palamiuc et al., 2017; Song and Avery, 2012; Sze et al., 2000; Waggoner et al., 1998)…" The referenced articles do not provide any data to substantiate the claim cited above. Palamiuc et al., 2017 reported that the lack of flp-7gene did not change feeding behaviour, locomotor activity and reproduction. However, the data on the effects of ASI neuron-specific flp-7 overexpression on the above-mentioned phenotypes were missing in the study of Palamiuc et al. and the authors do not show any data on 5-HT signalling regulating other phenotypes in flp-7^tg^ line apart from fat loss. The authors should provide appropriate citations and specify the phenotypes, or tone down statements and complete the phenotyping with follow up studies. The authors show that both 5-HT and FLP-7 induced fat loss was partially repressed in atfs-1 null mutants as shown in Figure 3A-B and Figure 3—figure supplement 1A. However, the data indicates a marked difference in fat loss following FLP-7 and 5-HT stimulation in this line. While the suppression of fat loss was partial in the atfs-1 null mutants treated with 5-HT (Figure 3—figure supplement 1A),the FlP-7 induced fat loss was completely recovered in the atfs-1 null mutants (Figure 3B). The authors could provide possible explanations for such discrepancy.

In Palamiuc et al., 2017, we showed that *flp-7* null mutants suppress 5-HT-induced fat loss, but do not suppress the other phenotypes associated with increased 5-HT signaling. We now add new data in Figure 1J, K and L showing that the *flp-7^tg^* line, as expected, does not display other predominant 5-HTergic phenotypes, including feeding, egg-laying, and locomotor responses on and off food. We have amended the text to reflect these additions.

Regarding the incomplete nature of *atfs-1* suppression with 5-HT, compared to *flp-7^tg^* animals. We have previously published that 5-HT elicits a dose-dependent stimulation of fat loss (Srinivasan et al., 2008). At a higher dose (5mM on the bacterial plate, used here), 5-HTergic phenotypes are clearly visible, and we observe 70-80% reduction in fat content. The advantage of this high dose is that it is a stringent means to identify suppressors. However, the caveat is that we rarely observe complete suppression at this dose, possibly because of non-specific effects. On the other hand, the *flp-7^tg^* line is a more physiological means to elicit fat loss (in the ~50% range) without altering other 5-HTergic phenotypes. In this setting, we observe more robust, complete suppression upon *atfs-1* removal. We have added a statement to this effect in the text.

3) In Figure 6C, the data from flp-7^tg^ and flp-7^tg^;atfs-1 are missing in both vector and atgl-1 RNAi conditions making difficult the interpretation of these results ¬- if the authors do not have these controls readily available then they should state this caveat, and provide the data in a follow up publication in bioRxiv or similar.

In Figure 6C, we had erroneously mislabeled the color of the bar graph for data from *flp-7^tg^* and *flp-7^tg^;atfs-1* in vector and *atgl-1* RNAi conditions, although we had included it. We are thankful to the reviewers for calling attention to this point, and have now corrected the colors on the bar graph, still in Figure 6C. With this correction, the data from these two lines are no longer missing.

4) "In contrast, hlh-11;atfs-1 double mutants resembled hlh-11 mutants alone, thus during increased FLP-7 secretion, atfs-1-dependent suppression of atgl-1 induction requires hlh-11 repression"a) In flp-7^tg^ line, atgl-1 expression significantly reduced in the absence of atfs-1 (Figures 6F, 3D-E) indicating that atfs1 activates atgl-1 during increased FLP-7 secretion. Also, atfs-1 mediated suppression of hlh-11 would induce atgl-1 expression (Figure 6D). Therefore, the above statement does not match the observations and should be rephrased.

We thank the reviewer, and have re-worded the statement to now correctly match the observations: “In contrast, *hlh-11;atfs-1* double mutants resembled *hlh-11* mutants alone, thus during increased FLP-7 secretion, *atfs-1-*dependent induction of *atgl-1* requires *hlh-11* repression.”

b) In general, the comprehensibility of Figure 6F would be greatly increased by adding representative fluorescent images as has been done with the preceding figures if the authors do not have these images readily available then they should state this caveat, and provide the data in a follow up publication in bioRxiv or similar.

We have now added the requested fluorescent images for the old Figure 6F, which has now been split into a new Figure 7, please see panels B and C.

5) Figure 7A and B show that during increased FLP-7 secretion, in the absence of both hlh-11 and atfs-1, flp-7^tg^ line lost almost half of their fat content although both lines have comparable level of atgl-1 expression (Figure 6G, flp-7^tg^ vs. flp-7^tg^;hlh-11;atfs-1). The authors could provide an explanation for these observations. Perhaps other signaling molecules might also be involved for fat loss in this model?

We have added a note to reflect this point in the section describing old Figure 7A, B, now new Figure 8A, B.

6) The authors normalize oxygen consumption in Figure 1H to worm rather than a measure of biomass such as protein. Does this mean that the worms are all the same size and all have the same biomass?

Throughout our studies, we have taken great care to ensure that for all genotypes and conditions tested, animals are of precisely the same age and developmental stage, in part because we have observed that fat metabolism is subject to great variation at different stages and days, and in part to ensure standard experimental rigor. Given our experimental context, we believe that OCR measured on a per worm basis, rather than normalized to protein is the more accurate method to determine physiological state.

7) Do the flp-7^tg^, 5-HT treated animals, hlh-11 mutant or the hlh-11^ox^ line show any alterations in fatty acid synthesis in parallel to the changes in OCR shown in Figures 1H, 4F and 5D?

Regarding fat synthesis changes in *flp-7^tg^*, 5-HT, *hlh-11* or HLH-11^OX^ mutants. In unpublished studies, we previously found that the major fat synthesis genes *fasn-1* (fatty acid synthase) and *pod-2* (acyl CoA carboxylase) were not transcriptionally altered in *flp-7^tg^* animals, or upon 5-HT treatment. Other groups had reported no change in fat synthesis in *tph-1* mutants, which lack 5-HT (Marc van Gilst, personal correspondence). We examined an approximately 5kb promoter region upstream of *pod-2* and *fasn-1*, and did not observe any HLH-11 cis-binding sites. Thus, the *c*urrent evidence does not support a role for fat synthesis in this pathway, although we cannot definitely exclude it as a contributing factor.

8) From the model presented in Figure 3F, the authors imply that the induction of atgl-1 is dependent upon the stress responses activated by atfs-1 in an indirect manner. In line with this, they showed atfs-1 induced repression of HLH-11 leads to atgl-1 activation (Figure 5). However, the authors did not provide any data that could connect hsp-60 stress response to atgl-1 induction ¬- if the authors do not have this readily available then they should state this caveat, and provide the data in a follow up publication in bioRxiv or similar.

We agree that data connecting *hsp-60* stress response to *atgl-1* induction is a critical aspect of the model. We did, in fact, provide this data in the original manuscript; please see Figure 2D-G.

9) Age is a massive effector of fat mass accumulation. Are the mutants flp-7^tg^, hlh-11, and atfs-1 or any combination therein developmentally altered to confound the interpretation of fat mass in Figure 7A-B?

As noted in point 6 above, for all mutants/genotypes and conditions tested, care was taken to ensure that only animals of the same age and stage (Day 1 of adulthood, within a 2-hour window) were compared and tested. We did not detect any developmental asynchrony or delay in any of the conditions tested.

Revisions expected in follow-up work:As stated in points #3, 4b, and 8 above.

We have either conducted new experiments, or provided data we had already collected and have included them in this revision, as noted in the points above.